# DYNAMIC CONTRASTIVE SKILL LEARNING WITH STATE-TRANSITION BASED SKILL CLUSTERING AND DYNAMIC LENGTH ADJUSTMENT

**Jinwoo Choi, Seung-Woo Seo**[*]
Department of Electrical and Computer Engineering
Seoul National University
Seoul, South Korea
{wlsdn9350,sseo}@snu.ac.kr

## ABSTRACT

Reinforcement learning (RL) has made significant progress in various domains, but scaling it to long-horizon tasks with complex decision-making remains challenging. Skill learning attempts to address this by abstracting actions into higher-level behaviors. However, current approaches often fail to recognize semantically similar behaviors as the same skill and use fixed skill lengths, limiting flexibility and generalization. To address this, we propose Dynamic Contrastive Skill Learning (DCSL), a novel framework that redefines skill representation and learning. DCSL introduces three key ideas: state-transition based skill representation, skill similarity function learning, and dynamic skill length adjustment. By focusing on state transitions and leveraging contrastive learning, DCSL effectively captures the semantic context of behaviors and adapts skill lengths to match the appropriate temporal extent of behaviors. Our approach enables more flexible and adaptive skill extraction, particularly in complex or noisy datasets, and demonstrates competitive performance compared to existing methods in task completion and efficiency.

## 1 INTRODUCTION

Reinforcement learning (RL) has advanced significantly in various domains (Vinyals et al. (2019), Silver et al. (2016)), yet scaling it to long-horizon tasks with complex decision-making remains challenging (Dulac-Arnold et al. (2019)). Skill learning from datasets ((Lynch et al. (2020), Sharma et al. (2019), Freed et al. (2023), Shi et al. (2022), Hao et al. (2024)) addresses these challenges by abstracting action sequences into higher-level behaviors, simplifying decision-making, and improving efficiency in long-horizon tasks.

Despite its potential, prior supervised skill-learning methods (Freed et al. (2023), Shi et al. (2022)) face significant limitations. First, action sequence encoding methods often struggle to identify semantically similar behaviors as the same skill when there are variations in the action sequences. As illustrated in Fig. 1, the actions required for 'grabbing the object' differ based on the object's position, leading to the learning of these behaviors as distinct skills. This limitation requires a large skill dimension space to accommodate various skills (Pertsch et al. (2021)), which in turn creates inefficiency when searching for the optimal combination of skills for new tasks. Since skills are stored without considering their semantic context, if a small skill dimension space is given, a dimensional collapse (Jing et al. (2021)) may occur where different behaviors are mapped to similar representations in the skill space.

Second, fixed skill lengths fail to reflect the varying durations of real-world behaviors. For instance, in pick-drop and pick-move data, if the length of the 'pick' behavior is shorter than the fixed skill length, the model fails to learn pick, drop, and move as individual skills, as humans would. While

---

[*]Corresponding author

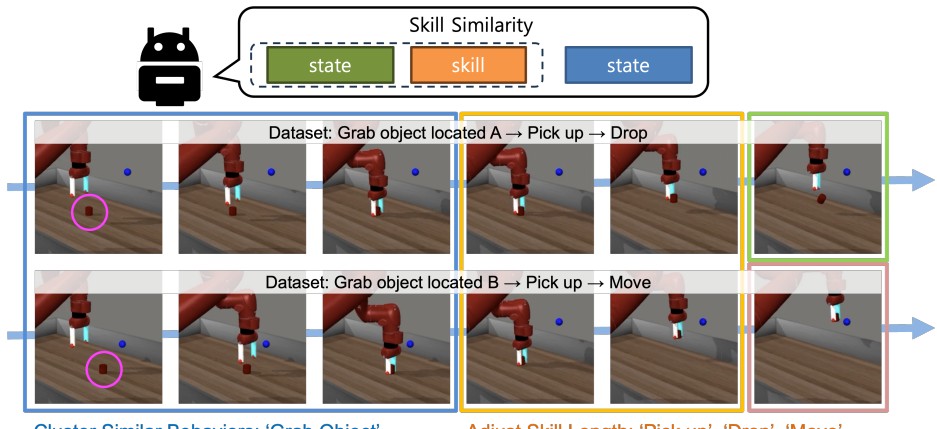

Figure 1: Illustration of DCSL's key ideas. Previous methods recognized 'grab object' as different skills based on object position, while DCSL clusters these into a unified skill using learned skill similarity. DCSL overcomes the limitation of fixed-length skills by dynamically adjusting skill lengths, enabling the recognition of shorter actions like 'Pick up' as independent skills.

fixed skill lengths have limitations, prior research using datasets of task-agnostic but meaningful behaviors has shown that learned skills can still retain some semantic content. However, this approach struggles with more complex or noisy data, where irrelevant or meaningless actions are included. In such scenarios, the ability to adapt skill lengths becomes crucial.

To address these challenges, we propose Dynamic Contrastive Skill Learning (DCSL), a novel framework that offers a new perspective on skill representation. DCSL introduces three key ideas: *state-transition based skill representation*, *skill similarity function learning*, and *dynamic skill length adjustment*. Unlike traditional methods that define skills based on action sequences, DCSL focuses on state transitions, enabling a more effective capture of the semantic context of behaviors. Through contrastive learning (Oord et al. (2018)), DCSL learns a skill similarity function that clusters semantically similar behaviors into the same skill, promoting more meaningful skill representations. DCSL leverages this function to dynamically adjust skill lengths based on observed state transitions, allowing for more flexible and adaptive skill extraction, particularly in noisy datasets. By combining these approaches, DCSL overcomes the limitations of existing methods, enabling effective skill learning that adapts to diverse environments and task structures.

In summary, our framework redefines skill representation and learning through three key contributions:

- We propose a state-transition based skill representation that captures the semantic context of behaviors more effectively than action sequence encoding. This approach enables the identification of similar skills even when the underlying action sequences differ.

- We introduce a skill similarity function learned through contrastive learning, which clusters semantically similar behaviors into the similar skill embeddings. This method effectively differentiates between states belonging to the same skill and those that do not.

- We develop a dynamic skill length adjustment method based on observed state transitions. Using the learned skill similarity function, our approach identifies high-similarity state transitions within the dataset and relabels skill lengths accordingly, allowing for more flexible and adaptive skill extraction in complex or noisy datasets.

## 2 RELATED WORK

**Skill Learning from Dataset**  Skill learning from datasets has emerged as a promising approach to improve efficiency and generalization in long-horizon reinforcement learning tasks. Foundational works like Pertsch et al. (2021) learned latent skill spaces from offline datasets, showing significant improvements in guiding exploration and task completion in novel environments. Lynch et al.

(2020) leveraged unlabeled play data to learn goal-conditioned policies in a latent plan space, while Pertsch et al. (2022) explored cross-domain skill transfer. In multi-task learning, Yoo et al. (2022) applied skill regularization and data augmentation to handle diverse offline datasets. Model-based approaches (Shi et al. (2022), Freed et al. (2023)) integrated skill dynamics with planning to improve long-term task planning without additional interaction. Existing skill learning methods lack flexibility due to their use of fixed-length action sequences, and struggle to recognize semantically similar behaviors. To address these limitations, our framework leverages contrastive learning to define skills based on state transitions and introduces a technique for dynamically adjusting skill lengths, enabling more flexible and adaptive skill extraction.

**Contrastive Learning in Reinforcement Learning**  Contrastive learning has gained significant traction in RL for enhancing state representations and improving sample efficiency. These methods typically employ contrastive losses to prioritize task-relevant features while disregarding irrelevant details. In the context of general state representation learning, Laskin et al. (2020) incorporated contrastive learning with off-policy RL algorithms, leveraging data augmentation and contrastive losses to extract meaningful representations from raw pixel inputs. For goal-conditioned RL, several works (Eysenbach et al. (2020), Eysenbach et al. (2022), Zheng et al. (2023)) have reframed goal achievement as a density estimation problem, using classifiers to predict future state distributions without explicit reward functions. In the domain of unsupervised skill discovery, Yang et al. (2023) applied multi-view contrastive learning to foster greater similarity within the same skill while maintaining diversity across different skills. Our DCSL framework builds upon these ideas by applying contrastive learning specifically to skill extraction from datasets. Unlike previous methods, we uniquely integrate dynamic skill length adjustment using learned contrastive representations, addressing the limitations of fixed-length approaches and enhancing adaptability across diverse tasks.

**Dynamic Length Adjustment in Temporal Abstractions**  Dynamic length adjustment in temporal abstractions enhances RL by enabling more flexible and hierarchical decision-making across variable time scales. The Options framework (Sutton et al. (1999)) laid the groundwork for temporal abstraction in reinforcement learning. Building on this, Bacon et al. (2017) proposed the Option-Critic architecture, which learns options end-to-end with differentiable option policies and termination functions. Shankar & Gupta (2020) introduced a method using temporal variational inference to discover skills from demonstrations in an unsupervised manner. Further contributions include the Harutyunyan et al. (2019), which learns option termination conditions, and the Jiang et al. (2022), which addresses the underspecification problem in skill learning by combining maximum likelihood objectives with a penalty on skill description length. While these approaches have significantly advanced temporal abstraction in RL, DCSL introduces a novel perspective. Instead of relying on learned termination functions, DCSL employs a skill similarity measure to dynamically adjust skill lengths. This approach allows DCSL to capture more meaningful and adaptable skill representations.

## 3 Problem Formulation and Preliminaries

### 3.1 Problem Formulation

In this work, we consider skill learning from unlabeled offline data, where the agent has access to a large, task-agnostic dataset $\mathcal{D} = \{\tau_1, \tau_2, \ldots, \tau_N\}$. Each trajectory $\tau_i = \{(s_t, a_t)\}_{t=1}^{T}$ consists of sequences of states $s_t \in \mathcal{S}$ and actions $a_t \in \mathcal{A}$, without any task-specific reward labels. We define a skill as a latent vector $z \in \mathcal{Z}$ that encodes a temporally extended behavior, capable of generating a sequence of actions given an initial state. Our aim is to learn skills that encapsulate diverse behaviors, which can later be adapted to solve new tasks efficiently. Each task is modeled as a Markov decision process (MDP) defined by the tuple $\{\mathcal{S}, \mathcal{A}, \mathcal{P}, r, \rho, \gamma\}$, representing states, actions, transition probabilities, rewards, initial state distribution, and discount factor.

### 3.2 Skill-based RL

Skill-based RL aims to improve sample efficiency and generalization in long-horizon tasks by abstracting low-level actions into higher-level skills. SPiRL (Pertsch et al. (2021)) is a framework that leverages prior data to learn a skill embedding space and a skill prior. SPiRL consists of three main components: a skill encoder $q(z|\mathbf{a})$, a skill decoder $\pi(a|s, z)$, and a skill prior $p_a(z|s)$. Here, bolded

**a** denotes a sequence of actions $\{a_0, \cdots, a_{H-1}\}$, where $H$ is a skill length. The skill encoder and skill decoder are trained through a Variational Autoencoder (VAE), while the skill prior is learned by modeling the distribution of $z$ from the data. During downstream task learning, the skill prior guides exploration by regularizing the policy towards skills that are likely under the prior with a weighting coefficient $\alpha$:

$$\pi^*(z|s) = \arg\max_\pi \mathbb{E}_\pi \left[ \sum_{t=0}^{T} r(s_t, a_t) - \alpha \cdot D_{\text{KL}}(\pi(z|s) \| p_a(z|s)) \right]. \tag{1}$$

This approach enables efficient learning of complex behaviors by leveraging the structure of previously seen tasks encoded in the skill prior.

### 3.3 Contrastive Learning for Representation Learning

Contrastive learning is a self-supervised technique that learns discriminative representations by contrasting positive and negative sample pairs. Initially developed for computer vision (Chen et al. (2020), He et al. (2020)) and natural language processing (Radford et al. (2021)), it has gained traction in RL for its ability to extract useful features from unlabeled data (Eysenbach et al. (2022)), Zheng et al. (2023)). The objective is to learn representations that bring positive (similar) samples closer together and push negative (dissimilar) samples apart.

In this framework, given two inputs, $u$ (anchor) and $v$ (positive or negative), the model maximizes the similarity between $u$ and positive samples $v^+$ while minimizing the similarity with negative samples $v^-$. The contrastive loss function (Ma & Collins (2018)) is:

$$\mathcal{L}_{\text{contrastive}} = \mathbb{E}_{(u,v^+)} \left[ \log \sigma(f(u, v^+)) \right] + \mathbb{E}_{(u,v^-)} \left[ \log(1 - \sigma(f(u, v^-))) \right], \tag{2}$$

where $f(u, v)$ is a similarity function, and $\sigma(\cdot)$ is the sigmoid function. Positive pairs $(u, v^+)$ are drawn from the joint distribution, while negative pairs $(u, v^-)$ are sampled from the product of marginals.

## 4 Proposed Method

This section introduces the Dynamic Contrastive Skill Learning (DCSL) framework, explaining its architecture and core components. DCSL aims to cluster semantically similar behaviors as the same skill and adjust skill lengths dynamically. The framework consists of three main components: skill learning through contrastive learning, skill length relabeling, and downstream task learning. An overview of the entire DCSL framework is presented in Fig. 2. The following subsections explore each component in depth.

### 4.1 Extracting Skills with Contrastive Learning

This section explains the process of skill learning using contrastive learning. Our goal is to learn skills such that similar behaviors are clustered into a single skill. First, we label an initial skill length $H_t$ for each $\{(s_t, a_t)\}_{t=1}^T$ in the given dataset $\mathcal{D}$. This is defined as the length of the skill when the initial state of the skill is $s_t$. The initial skill lengths are set to the same value.

**Skill Embedding** We select four key states from each skill duration: the initial state $s_t$, the terminal state $s_{t+H_t-1}$, and two randomly sampled intermediate states $s_{t+a}$ and $s_{t+b}$ (where $t < t + a < t + b < t + H_t - 1$). This approach provides a snapshot of the skill's progression in a unified representation, enabling a more effective representation of the overall trajectory and behavioral context. The selected states $\vec{s}_t = \{s_t, s_{t+a}, s_{t+b}, s_{t+H_t-1}\}$ are encoded into a skill embedding $z_t$ using an LSTM-based encoder $q_{\theta_q}(z|\vec{s})$. The loss function for skill embedding is:

$$\mathcal{L}_{\text{embedding}} = \mathbb{E}_{(\vec{s},\vec{a})\sim\mathcal{D}}[\mathbb{E}_{q_{\theta_q}(z|\vec{s})}[\lambda_{\text{BC}} \cdot \log \pi_{\theta_\pi}(\vec{a}|\vec{s}, z)] - \beta \cdot D_{\text{KL}}(q_{\theta_q}(z|\vec{s})|p(\mathbf{z}))$$
$$+ \lambda_{\text{SP}} \cdot D_{\text{KL}}(\mathbf{sg}(q_{\theta_q}(z|\vec{s}))|p_{\theta_p}(z|s))]. \tag{3}$$

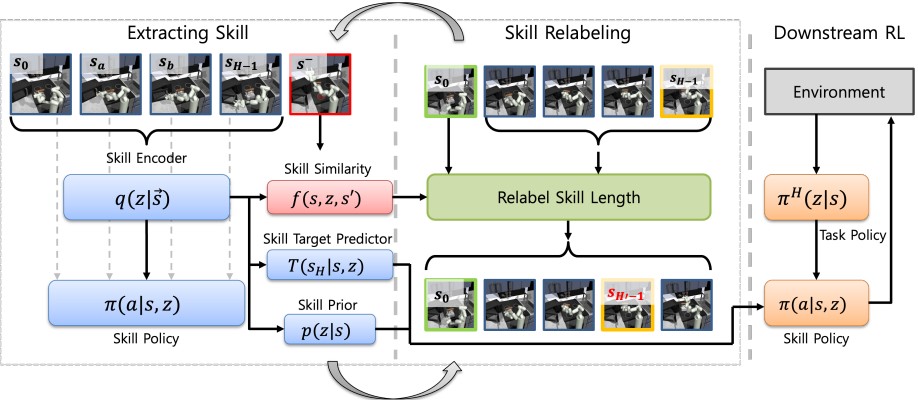

Figure 2: The overall architecture of our framework, including skill extraction, skill length relabeling, and the application of skills to downstream tasks. Our method samples four states to encode skills, with red-bordered states representing negative samples. In the relabeling process, green borders indicate the skill initial state, and yellow borders denote the skill terminate state. The learned skills are then utilized in downstream RL tasks.

Here, $\vec{a}$ represents the actions corresponding to $\vec{s}$, while $\lambda_{\text{BC}}$, $\lambda_{\text{SP}}$, and $\beta$ are weighting coefficients, and $\theta_q$, $\theta_\pi$, and $\theta_p$ are trainable parameters. Additionally, **sg** denotes the stop gradient operator. The prior $p(\mathbf{z})$ is defined as a tanh-transformed standard Gaussian distribution, represented as $\tanh(\mathcal{N}(0,1))$. The skill decoder $\pi(\vec{a}|\vec{s}, z)$ reconstructs these actions using behavior cloning loss. The skill prior $p(z|s_t)$ captures the distribution of likely skills for a given state.

**Contrastive Learning for Skill Discrimination**   Our main goal is to cluster similar behaviors into the same skill. To achieve this, we introduce a skill similarity function $f_{\theta_f}(s, z, s') = \langle \phi_{\theta_\phi}(s, z), \psi_{\theta_\psi}(s') \rangle$. Here, $\phi_{\theta_\phi}(s, z)$ is a skill-conditioned state representation, and $\psi_{\theta_\psi}(s')$ is a state representation, both of which are trainable networks (Eysenbach et al. (2022)). Our core assumption is that semantically similar behaviors will display similar state change patterns after applying a skill. To train the skill similarity function, we need both positive and negative samples. Positive samples pair the initial state $s_t$ with the skill $z_t$ and an intermediate state $s_{t+b}$, chosen to capture long-term skill effects. Negative samples are states $s^-$ that are unreachable from $s_t$ given the skill $z_t$, sampled from different skill trajectories. This can be expressed as:

$$s^- \sim \text{Uniform}(\{s \in \mathcal{D} | \exists z' \neq z, s \in \tau^{skill}(z')\}), \tag{4}$$

where $\tau^{skill}(z) = \{s_t, \ldots, s_{t+H_t-1}\}$ represents the set of states belonging to the trajectory of skill $z$. For simplicity, we denote this distribution as $p(s|z' \neq z)$. We use the Noise Contrastive Estimation (NCE) binary loss function (Ma & Collins (2018)) to train the skill similarity function. Here, $\sigma(\cdot)$ denotes the sigmoid function, while $\lambda_{\text{CL}}$ represents the weighting coefficient, and $\theta_f$ denotes the trainable parameter. The contrastive loss is as follows (a detailed analysis can be found in Appendix A.1.):

$$\mathcal{L}_{\text{contrastive}} = \lambda_{\text{CL}} \cdot \mathbb{E}_{(\vec{s},\vec{a}) \sim \mathcal{D}} \left[ \mathbb{E}_{q_{\theta_q}(z|\vec{s})} \left[ -\log \sigma(f_{\theta_f}(s, z, s_b)) \right. \right.$$
$$\left. \left. - \mathbb{E}_{s^- \sim p(s|z' \neq z)} \left[ \log(1 - \sigma(f_{\theta_f}(s, z, s^-))) \right] \right] \right]. \tag{5}$$

**Skill Terminate State Predictor**   For downstream task learning, determining the point to start the next skill is crucial due to the variable lengths of skills in our method. Inspired by SkiMo (Shi et al. (2022)), we repurposed their dynamics module to function as a skill target state predictor specifically for use in downstream tasks. This predictor is capable of handling these variable skill lengths. Our approach predicts the skill target state $s_{t+H_t}$, which is the state after the skill has been executed from the initial state $s_t$. When the skill target state is reached during downstream task execution, the skill is considered complete, allowing the next skill to proceed.

We use a state encoder $E_{\theta_E}(s_t)$ and observation decoder $O_{\theta_O}(E_{\theta_E}(s_t))$ to map raw observations to a latent state space and ensure accurate state representations by reconstructing the original observations. The loss function for the skill target state predictor $T_{\theta_T}(\cdot)$ is as follows:

$$
\mathcal{L}_{\text{target}} = \sum_t \mathbb{E}_{\vec{s} \sim \mathcal{D}} \Big[ \mathbb{E}_{z_t \sim q_{\theta_q}(z_t | \vec{s})} \Big[ \lambda_{\text{RE}} \| s_t - O_{\theta_O}(E_{\theta_E}(s_t)) \|^2
$$
$$
+ \lambda_{\text{ST}} \| T_{\theta_T}(E_{\theta_E}(s_t), z_t) - E_{\theta_E}(s_{t+H_t}) \|^2 \Big] \Big]. \tag{6}
$$

Here, $\lambda_{\text{RE}}$ and $\lambda_{\text{ST}}$ represent the weighting coefficients, and $\theta_O, \theta_E, \theta_T$ are the trainable parameters. This loss includes the observation reconstruction term $\| s_t - O_{\theta_O}(E_{\theta_E}(s_t)) \|^2$ and the skill target state prediction term $\| T_{\theta_T}(E_{\theta_E}(s_t), z_t) - E_{\theta_E}(s_{t+H_t}) \|^2$.

**Objective Function** The total objective function for the model is a combination of the skill embedding, contrastive learning, and skill target state predictor losses:

$$
\mathcal{L}_{\text{total}} = \mathcal{L}_{\text{embedding}} + \mathcal{L}_{\text{contrastive}} + \mathcal{L}_{\text{target}}. \tag{7}
$$

### 4.2 SKILL LENGTH RELABELING

We introduce a dynamic skill length relabeling process based on the learned skill similarity function $f_{\theta_f}(s, z, s')$. Starting from each skill's initial state $s_t$ in the dataset, we obtain $z_t$ using the pre-relabeling skill length $H_t$. We then compute similarity values for subsequent states $s_{t+\alpha}$ ($\alpha > 0$) and define the new skill length as the maximum number of consecutive time steps where the similarity remains above a threshold $\epsilon$:

$$
H_t' = 1 + \max \left\{ \alpha \,\middle|\, f_{\theta_f}(s_t, z_t, s_{t+\alpha}) > \epsilon \right\}. \tag{8}
$$

This approach allows for dynamic adjustment of skill lengths based on observed state transitions, ensuring that each skill's duration accurately reflects the temporal extent of its effect. To ensure stable learning, relabeling is performed every $T_{\text{relabel}}$ training steps. Additionally, constraints on skill length ($\delta_{\min} \le H_t \le \delta_{\max}$) are applied to prevent abrupt changes. A detailed convergence analysis and theoretical justification are provided in Appendix A.2.

### 4.3 DOWNSTREAM TASKS LEARNING

The learned skills can be applied to downstream tasks using both model-free and model-based approaches. As our research focuses on skill extraction and representation, we utilized previous approaches such as SPiRL (Pertsch et al. (2021)) and SkiMo (Shi et al. (2022)) for downstream learning. In the model-free setting, Soft Actor-Critic (SAC) (Haarnoja et al. (2018)) is used to train a high-level policy that selects skills based on the current state. Meanwhile, in the model-based setting, skill target state prediction is combined with the Cross-Entropy Method (CEM) (Rubinstein (1997)) to enable long-horizon planning through skill simulations.

A key feature of our framework in downstream learning is the dynamic adjustment of skill durations during execution. Unlike previous methods with fixed-length skills, our approach uses a skill target state predictor to adapt skill execution. Skills are continued to execute until the predicted target state is reached or until a certain number of timesteps, at which point the high-level policy selects the next skill. Detailed explanations of the model-free and model-based implementations are provided in Appendix B.2.

## 5 EXPERIMENTS

We designed our experiments to answer the following three key questions: (1) Can the skills learned through DCSL capture meaningful behaviors that enable efficient performance on new tasks in both

navigation and manipulation domains? (2) Is DCSL able to extract meaningful and common behaviors through skill length adjustment even in more complex or noisy data that include irrelevant or meaningless actions? (3) Are similar behaviors effectively clustered into similar skills, resulting in an efficiently learned skill representation space? Our experimental results provide compelling evidence to address these questions, demonstrating the effectiveness of DCSL across various environments and data conditions.

## 5.1 EXPERIMENTAL SETUP

We conducted experiments in the antmaze, kitchen, and pick-and-place environments (Fig.6). Antmaze is a long-horizon navigation task, where the goal is for an Ant agent to reach a designated goal. We employed two variants, Antmaze-Medium and Antmaze-Large. For each environment, we used the 'antmaze-medium-diverse' and 'antmaze-large-diverse' datasets provided by D4RL (Fu et al. (2020)). In the Kitchen environment, the agent is tasked with completing four independent subtasks. We utilized the 'kitchen-mixed' dataset from D4RL for this environment. In the Meta-world Pick-and-Place environment (Yu et al. (2020)), we used datasets provided by Yoo et al. (2022) containing varying levels of noise. These datasets are ordered from least to most noisy: Medium-Expert (ME), Medium-Replay (MR), and Replay (RP).

**Baselines** To assess the effectiveness of our framework, we compared it with several well-established baselines. First, to evaluate the utility of the skills learned by DCSL, we used two skill learning comparison groups: SPiRL (Pertsch et al. (2021)) and SkiMo (Shi et al. (2022)). SPiRL employs a learned skill prior to guide exploration, while SkiMo integrates skill dynamics with model-based planning. Both methods defined skills as fixed-length action sequences. We evaluated SkiMo in two variants: SkiMo-SAC and SkiMo-CEM, using the SAC algorithm and CEM respectively. Our research focuses primarily on the skill extraction process. To evaluate our extracted skills, we applied existing methods for downstream task learning, testing two versions of our DCSL framework: Ours-SAC and Ours-CEM. In all experimental environments, the predefined skill length for the baseline methods and the initial skill length for DCSL were set to 10. To constrain the skill space, we set the skill dimension to 5 for Antmaze and Kitchen environments, and 2 for Pick-and-Place. To further evaluate DCSL's effectiveness in utilizing offline datasets for task performance, we included additional baselines. We compared against Behavioral Cloning (BC) (Pomerleau (1988)), which learns policies directly from demonstration data, and Conservative Q-Learning (CQL) (Kumar et al. (2020)), an offline RL method that ensures policy actions remain within the data distribution. We also included CQL+Off-DADS (Sharma et al. (2020)), which learns skills by maximizing mutual information between skills and trajectories, and CQL+OPAL (Ajay et al. (2020)), which uses unsupervised learning to discover skills that reduce temporal abstraction. While CQL+Off-DADS and CQL+OPAL also learn skills, they differ from DCSL in their use of CQL for high-level policy learning.

## 5.2 EXPERIMENTAL RESULTS

Fig. 3 and Table 1 present the results for each environment through learning graphs and success rates, while Table 2 shows the number of timesteps required to complete the tasks. These results demonstrate the performance and efficiency of our DCSL framework across various environments and task complexities.

**Antmaze-medium** The Antmaze environment is a sparse reward task where the agent only receives a reward upon reaching the goal. As a result, it requires extensive maze exploration, with skill learning efficiency dependent on the clustering of similar behaviors. Effective clustering offers useful skill representation, redundancy reduction, and improved generalization. Results from Antmaze-Medium and Antmaze-Large demonstrate DCSL's effective skill clustering and representation capabilities. In Antmaze-Medium, Ours-SAC outperformed SkiMo-SAC, suggesting DCSL generated more accurate skill representations, leading to improved exploration efficiency. SkiMo-CEM showed high success rates in this environment, while Ours-CEM struggled to achieve significant performance. This discrepancy can be attributed to DCSL's variable skill lengths, which destabilize CEM's planning horizon. The inconsistent time scales hinder CEM's effectiveness in long-horizon tasks, exacerbating cumulative errors. Comparing with other baselines utilizing offline datasets, we observe that BC and CQL, which do not employ skill learning, struggle to reach

Table 1: Comparison of success rates. Boxes containing '-' denote results that could not be obtained due to the unavailability of a public implementation code.

| Environment | Method | | | | |
|---|---|---|---|---|---|
| | BC | CQL | CQL+Off-DADS | CQL+OPAL | Ours-SAC |
| Ant-Medium | 0.0 | $53.7 \pm 6.1$ | $59.6 \pm 2.9$ | $\mathbf{81.1} \pm 3.1$ | $68.0 \pm 36.9$ |
| Ant-Large | 0.0 | $14.9 \pm 3.2$ | - | $70.3 \pm 2.9$ | $\mathbf{73.7} \pm 5.9$ |
| Kitchen | 47.5 | $52.4 \pm 2.5$ | - | $69.3 \pm 2.7$ | $\mathbf{94.7} \pm 1.5$ |

the goal. While DCSL outperforms CQL+Off-DADS, CQL+OPAL achieves the highest success rate. This performance difference can be attributed to the varying approaches in learning high-level policies, similar to what we observed with SkiMo-CEM.

**Antmaze-large** In Antmaze-Large, Ours-SAC significantly outperformed all other methods, demonstrating DCSL's generalization ability in more complex environments. The performance gap with SkiMo-SAC implies DCSL better addresses the sparse reward problem. CEM-based methods' performance degradation highlights DCSL's superior skill representation space learning in challenging environments. These results prove DCSL enhanced exploration efficiency and generalization ability across various environmental scales through effective skill clustering and representation. In comparison to other offline dataset baselines, BC and CQL face learning difficulties similar to those in Antmaze-medium. Although CQL+OPAL demonstrates high performance, DCSL achieves a higher success rate due to its superior exploration efficiency. This highlights DCSL's ability to effectively navigate more complex environments.

**Kitchen** Unlike the Antmaze environment, the Kitchen environment demands precise object manipulation skills. DCSL's performance here demonstrates its flexibility and adaptability. Ours-SAC successfully completed almost all four subtasks, indicating DCSL's effectiveness in tasks requiring delicate action sequences, despite using state-based skill embeddings. Our approach performed similarly or better than action sequence-based methods, suggesting DCSL better captures task essences by focusing on state transitions. While Ours-SAC and SkiMo-CEM completed a similar number of subtasks, our method required fewer timesteps to accomplish the tasks, as shown in Table 2. This efficiency highlights the effectiveness of the skills that DCSL has learned. Although Ours-CEM showed slightly lower performance, the overall results of DCSL are promising, particularly considering its ability to generalize across different task types. This underscores DCSL's strength in learning transferable skills that are effective in both exploration-centric tasks and those requiring precise manipulation, demonstrating robust performance across diverse environments. When compared to other offline dataset baselines, DCSL significantly outperforms its counterparts. The superior performance in this manipulation task, which requires precise control, demonstrates the effectiveness of state-transition based skills in capturing intricate action sequences. This result underscores DCSL's adaptability to tasks demanding detailed object manipulation.

**Pick-and-Place** The Pick-and-Place environment evaluates the impact of data quality on skill learning, showcasing the importance of DCSL's skill length adjustment. In the high-quality ME dataset, all methods achieved similar success rates. However, performance differences became more pronounced in the lower-quality MR dataset, with DCSL showing improved performance. In the noisy RP dataset, Ours-SAC outperformed all other variants, demonstrating DCSL's effectiveness in handling data with high noise and irrelevant actions. These results show that DCSL maintains strong performance across datasets of varying quality, with its importance becoming more evident as data quality decreases. In the MR and RP datasets, our method not only achieved high success rates but also required the least number of timesteps to complete the tasks.

## 5.3 ABLATION STUDY

The ablation study (Fig. 4) conducted in the Pick-and-Place environment utilized three datasets (ME, MR, and RP) to evaluate the impact of the similarity function and skill length readjustment process in the DCSL framework. In the ME dataset, all methods achieved similarly high success rates, suggesting that skill length readjustment has relatively little impact on high-quality data. However, without a similarity function, efficient skills cannot be effectively learned, resulting in the high-level policy requiring more training steps to converge. The benefits of skill length readjustment became

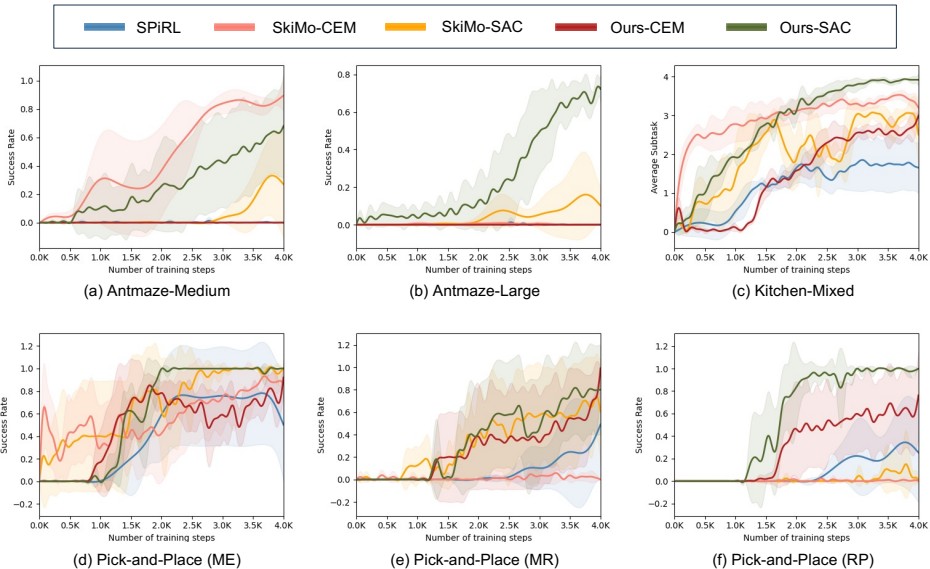

Figure 3: Performance comparison across Antmaze-Medium, Antmaze-Large, Kitchen, and Pick-and-Place environments (with 5 different random seeds). Results demonstrate the adaptability and effectiveness of DCSL in handling diverse and long-horizon tasks. Dark lines represent the average returns, and shaded areas represent standard deviations. A single training step was conducted after rolling out one episode.

Table 2: Comparison of the number of timesteps required to complete tasks across different environments and algorithms.

| Environment | SPiRL | SkiMo-CEM | SkiMo-SAC | Ours-CEM | Ours-SAC |
|---|---|---|---|---|---|
| | | | Method | | |
| Ant-Medium | $988.5 \pm 19.8$ | $\mathbf{311.2} \pm 95.7$ | $833.7 \pm 288.0$ | $1000.0 \pm 0.0$ | $453.6 \pm 143.7$ |
| Ant-Large | $990.2 \pm 19.5$ | $993.5 \pm 13.9$ | $881.5 \pm 164.9$ | $1000.0 \pm 0.0$ | $\mathbf{672.2} \pm 72.9$ |
| Kitchen | $276.6 \pm 5.9$ | $205.8 \pm 29.0$ | $251.3 \pm 23.7$ | $262.0 \pm 20.1$ | $\mathbf{165.1} \pm 4.4$ |
| PP (ME) | $87.8 \pm 65.0$ | $\mathbf{54.1} \pm 21.3$ | $58.0 \pm 6.8$ | $76.0 \pm 15.8$ | $80.1 \pm 13.7$ |
| PP (MR) | $138.0 \pm 63.2$ | $184.8 \pm 24.0$ | $87.3 \pm 57.6$ | $62.9 \pm 5.8$ | $\mathbf{56.1} \pm 5.1$ |
| PP (RP) | $130.6 \pm 69.4$ | $193.2 \pm 13.5$ | $200.0 \pm 0.0$ | $85.1 \pm 22.3$ | $\mathbf{64.4} \pm 16.3$ |

more apparent in the MR dataset. Ours-CEM outperformed 'w/o relabel (CEM)', indicating that skill length readjustment enhances the noise handling capability of CEM-based methods. The importance of skill length readjustment was most pronounced in the RP dataset. Ours-SAC achieved the highest success rate, while 'w/o relabel (CEM)' and 'w/o sim' showed significantly lower success rates. These results demonstrate that the similarity function and skill length readjustment process in DCSL plays a crucial role in maintaining consistently strong performance across datasets of varying quality.

## 5.4 QUALITATIVE EVALUATION OF SKILL CLUSTERING

We conducted a qualitative evaluation (Fig. 5) to assess whether DCSL effectively clustered similar behaviors into the same skill, thus efficiently learning the skill representation space. For this purpose, we modified the Antmaze environment by removing all obstacles except the borders and placing the Ant agent at the center of the maze. We sampled 50 random skills and visualized the Ant's movements during the execution of each skill.

The results show that for SPiRL and SkiMo, similar behavior patterns repeatedly appeared across multiple random skills. This suggests that these methods are experiencing a dimensional collapse, where similar behaviors are learned as different skills. In contrast, DCSL exhibited a wide range of behavior patterns, including movements in various directions. This demonstrates that DCSL effectively clustered similar behaviors, leading to an efficient learning of the skill representation

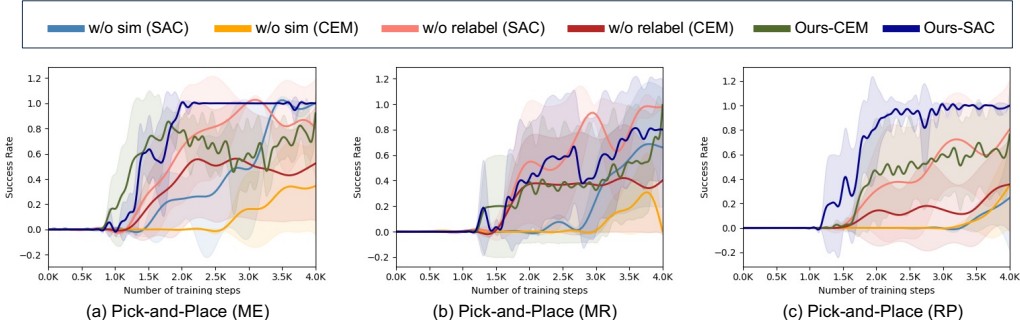

Figure 4: Ablation study results in the Pick-and-Place environment across the ME, MR, and RP datasets, comparing the impact of the similarity function and skill length relabeling in our framework. These results highlight the significance of these techniques in maintaining performance robustness in suboptimal and noisy environments.

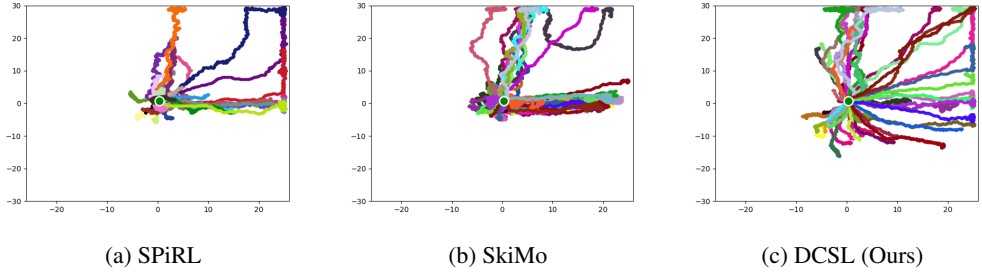

Figure 5: Comparison of execution results for 50 randomly sampled skills in the Antmaze environment. Each colored line represents the trajectory of the Ant agent during the execution of a single skill. While SPiRL and SkiMo show similar behavior patterns repeating across multiple skills, DCSL demonstrates movements in various directions and patterns. This suggests that DCSL has learned a more efficient skill representation space by effectively clustering similar behaviors.

space. These findings indicate that DCSL can learn a more diverse and meaningful set of skills compared to existing methods.

## 6 CONCLUSION

In this paper, we introduced Dynamic Contrastive Skill Learning (DCSL), a novel framework that addresses challenges in long-horizon reinforcement learning tasks by focusing on state-transition rather than fixed action sequences. DCSL makes two key contributions: it uses contrastive learning to cluster semantically similar behaviors, enabling more effective skill extraction, and it dynamically adjusts skill lengths based on a learned similarity function. Our experiments in navigation and manipulation tasks demonstrated DCSL's ability to extract useful skills from diverse datasets, including those containing a mix of relevant behaviors and irrelevant or random actions, enhancing generalization and adaptability across various environments and showing competitive performance compared to existing methods in task completion and efficiency.

Looking ahead, we envision several promising directions for future work. First, we aim to extend the use of the learned similarity function beyond skill learning and length adjustment to downstream task learning, potentially improving task performance by leveraging the semantic understanding of skills during execution. Second, we plan to explore the integration of language instructions with skill learning using contrastive methods, enabling agents to follow natural language commands and generate relevant skills, thus expanding the framework's potential in interactive environments. Lastly, we intend to investigate the application of DCSL in multi-agent systems and dynamic task structures, further testing its adaptability and scalability. These extensions would not only enhance the capabilities of DCSL but also bridge the gap between skill learning, language understanding, and task execution in reinforcement learning.

## REPRODUCIBILITY STATEMENT

We implemented all learning models using PyTorch. The experiments were conducted in environments provided by D4RL (Fu et al. (2020)) and Metaworld (Yu et al. (2020). A detailed explanation of our framework's overall algorithm, model implementation details, and environment configurations are provided in Appendix B and Appendix C.

## ACKNOWLEDGEMENTS

This research was supported by the Challengeable Future Defense Technology Research and Development Program through the Agency For Defense Development(ADD) funded by the Defense Acquisition Program Administration(DAPA) in 2024(No.915108201)

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

# A THEORETICAL ANALYSIS

## A.1 THEORETICAL ANALYSIS OF CONTRASTIVE LEARNING FOR SKILL DISCRIMINATION

This section provides a rigorous theoretical foundation for understanding how contrastive learning enables effective skill clustering in DCSL. We present two key theorems that demonstrate the relationship between our contrastive learning objective and mutual information maximization, as well as its connection to true skill discrimination.

### A.1.1 CONTRASTIVE LEARNING OBJECTIVE AND MUTUAL INFORMATION MAXIMIZATION

The contrastive learning objective used in DCSL is defined as:

$$\mathcal{L}_{\text{contrastive}} = \lambda_{\text{CL}} \cdot \mathbb{E}_{(\vec{s},\vec{a}) \sim \mathcal{D}} \left[ \mathbb{E}_{q_{\theta_q}(z|\vec{s})} \left[ -\log \sigma(f_{\theta_f}(s, z, s_b)) \right. \right.$$
$$\left. \left. - \mathbb{E}_{s^- \sim p(s|z' \neq z)} \left[ \log(1 - \sigma(f_{\theta_f}(s, z, s^-))) \right] \right] \right]. \quad (9)$$

where $f_{\theta_f}(s, z, s') = \langle \phi_{\theta_\phi}(s, z), \psi_{\theta_\psi}(s') \rangle$ is the skill similarity function.

**Theorem 1.** *Assuming an optimal discriminator, the contrastive learning objective maximizes the mutual information between skills and state transitions.*

*Proof.* The optimal solution for the inner expectation of the contrastive loss is achieved when:

$$\sigma(f_{\theta_f}(s, z, s')) = \frac{p(s'|s, z)}{p(s'|s, z) + p(s')}$$

Substituting this into the contrastive loss and simplifying:

$$\mathcal{L}^*_{\text{contrastive}} = -\mathbb{E}_{(s,z,s') \sim p(s,z,s')} \left[ \log \frac{p(s'|s, z)}{p(s')} \right] = -I(Z; S'|S)$$

Where $I(Z; S'|S)$ is the conditional mutual information between skills $Z$ and future states $S'$ given the current state $S$.

This result demonstrates that minimizing the contrastive loss is equivalent to maximizing the mutual information between skills and state transitions, leading to more discriminative skill representations. As a corollary, maximizing this mutual information promotes improved skill clustering. Intuitively, this can be explained by three key points: First, higher mutual information indicates that skills are more predictive of future states. Second, similar state transitions will be associated with similar skill representations to maximize this predictiveness. Finally, this similarity in skill representations for similar behaviors naturally leads to clustering in the skill space.

### A.1.2    LOWER BOUND ON SKILL DISCRIMINATION

**Theorem 2.** *The contrastive learning objective provides a lower bound on the true skill discrimination task.*

*Proof.* Let $C(s, z, s')$ be a binary random variable indicating whether $s'$ is reachable from $s$ using skill $z$. The true skill discrimination task can be formulated as maximizing:

$$\mathbb{E}_{(s,z,s')\sim p(s,z,s')}[\log p(C = 1|s, z, s')] + \mathbb{E}_{(s,z)\sim p(s,z),s'\sim p(s')}[\log p(C = 0|s, z, s')]$$

It can be shown that our contrastive loss provides a lower bound on this objective:

$$\mathcal{L}_{\text{contrastive}} \leq \mathbb{E}_{(s,z,s')\sim p(s,z,s')}[\log p(C = 1|s, z, s')] + \mathbb{E}_{(s,z)\sim p(s,z),s'\sim p(s')}[\log p(C = 0|s, z, s')]$$

This theorem demonstrates that by optimizing our contrastive loss, we are effectively learning to discriminate between reachable and unreachable states given a skill, which is crucial for meaningful skill clustering.

### A.1.3    CONCLUSION

These theoretical results provide a rigorous foundation for understanding why contrastive learning leads to effective skill clustering in DCSL. By maximizing the mutual information between skills and state transitions, and providing a lower bound on the true skill discrimination task, our method naturally groups similar behaviors into coherent skill representations.

## A.2    THEORETICAL ANALYSIS OF SKILL LENGTH RELABELING

This section provides an intuitive explanation and brief analysis illustrating how constraining skill lengths and performing periodic relabeling contribute to stabilizing the training process.

### A.2.1    REDUCING TRAINING INSTABILITY BY LIMITING SKILL LENGTH VARIATIONS

By enforcing constraints on skill lengths ($\delta_{\min} \leq H_t \leq \delta_{\max}$), we prevent the skill duration $H_t$ from changing abruptly during the relabeling process. Skills group the data into distinct behavioral categories, allowing the model to handle each category with specialized representations. Sudden large changes in skill lengths can cause abrupt shifts in the data distribution, potentially leading to unstable training dynamics. By limiting the variation in $H_t$, we ensure that changes in the data distribution occur gradually, allowing the model to adapt more smoothly.

Additionally, small changes in $H_t$ lead to gradual transitions in the loss landscape, resulting in more consistent gradient updates during backpropagation. This consistency helps stabilize the learning process by preventing drastic fluctuations in gradients.

#### ANALYSIS: BOUNDING CHANGES IN LOSS AND GRADIENTS

When the change in skill length $\Delta H_t = H_t' - H_t$ is limited by $|\Delta H_t| \leq \delta_{\max} - \delta_{\min}$, the variations in the loss function $\mathcal{L}_{\text{total}}$ and its gradients with respect to model parameters are expected to be bounded. While exact mathematical bounds depend on the specific properties of the loss function and model architecture, constraining skill lengths reduces the potential for large fluctuations in loss and gradients, thereby contributing to training stability.

### A.2.2    MODEL ADAPTATION THROUGH PERIODIC RELABELING

Performing skill length relabeling every $T$ training steps allows the model sufficient time to adapt to the current skill assignments before they are updated again. Reducing the frequency of relabeling prevents the model from continuously adjusting to new skill assignments, minimizing potential disruptions during training. This periodic approach enables the model parameters to be updated based on a consistent set of skills over multiple iterations, facilitating more stable convergence.

ANALYSIS: REDUCING GRADIENT VARIANCE

By maintaining consistent skill assignments over $T$ training steps, the variability in parameter updates can be reduced. While we do not assume that gradients are independent or identically distributed across steps, the consistent skill assignments help in smoothing out fluctuations in the training process. Averaging updates over multiple steps with stable skill assignments contributes to more consistent gradient directions and magnitudes, aiding in stable learning.

### A.2.3 ENSURING STABILITY THROUGH THE COMBINATION OF CONSTRAINTS AND PERIODICITY

Combining constraints on skill lengths with periodic updates ensures that changes to the model's input distribution and internal representations occur gradually. This gradual evolution minimizes abrupt shifts that could destabilize training. Stable skill lengths and periodic updates enable the model to effectively learn skill embeddings that capture semantically similar state transitions, enhancing representation learning.

ANALYSIS: SMOOTHING THE OPTIMIZATION LANDSCAPE

Constraining skill lengths and performing periodic relabeling help smooth the optimization landscape by preventing sudden changes in the loss function with respect to model parameters. This smoothing effect reduces the likelihood of encountering sharp gradients or irregularities that can hinder optimization. While we do not make specific assumptions about the second derivatives (Hessian) of the loss function, the general effect of these techniques is to promote a smoother loss surface, facilitating more reliable convergence of optimization algorithms.

## B  IMPLEMENTATION DETAILS

This section describes the implementation details of our framework.

### B.1  ALGORITHM

Algorithms 1 and 2 describe, respectively, the method for extracting skills from datasets and the process of relabeling skill lengths in our framework.

### B.2  DOWNSTREAM RL

Here, we outline the algorithm used to train downstream tasks. Rather than developing a novel approach for efficiently searching through skill combinations, our focus was on refining the skills themselves. To showcase the effectiveness of these learned skills, we used the same method for combining skills as in previous work, drawing on established approaches such as SPiRL (Pertsch et al. (2021)) and SkiMo (Shi et al. (2022)). A brief overview of each method is provided below, with further details available in the corresponding papers.

**Model-Free Approach**  For the model-free setting, we employ the SPiRLPertsch et al. (2021) framework, which introduces a skill prior to guide exploration in the skill space. This approach leverages a learned skill prior $p_a(z|s_t)$, which captures which skills are most relevant given the current state, reducing exploration inefficiencies.

The high-level policy $\pi_\theta(z|s_t)$ selects skill embeddings $z_t$ that are decoded into action sequences. To encourage the exploration of meaningful skills, we regularize the policy with the learned skill prior using the following objective:

$$L_{\text{SPiRL}} = \mathbb{E}_\pi \left[ \sum_{t=0}^{T/H} (r_t - \alpha D_{KL} (\pi_\theta(z_t|s_t) \| p_a(z|s_t))) \right] \tag{10}$$

---

**Algorithm 1** Skill Extraction with Contrastive Learning

---

**Require:** Unlabeled offline dataset $\mathcal{D} = \{\tau_1, \tau_2, \ldots, \tau_N\}$, where each $\tau_i = \{(s_t, a_t)\}_{t=1}^T$, Relabeling interval $T_{\text{relabel}}$
**Ensure:** Learned skill embeddings $z \in \mathcal{Z}$
  1: Initialize skill encoder $q_{\theta_q}$, skill decoder $\pi_{\theta_\pi}$, skill prior $p_{\theta_p}$
  2: Initialize skill similarity function $f_{\theta_f}$
  3: Initialize state encoder $E_{\theta_E}$, observation decoder $O_{\theta_O}$, skill termination predictor $T_{\theta_T}$
  4: Initialize skill length $H_t$ for each $(s_t, a_t)$ in $\mathcal{D}$
  5: **for** each training iteration $i$ **do**
  6:      Sample batch of skill trajectories $\{\tau_i^{skill}\}_{i=1}^B$ from $\mathcal{D}$
  7:      **for** each $\tau_i^{skill}$ in batch **do**
  8:         Select key states $\vec{s}_t = \{s_t, s_{t+a}, s_{t+b}, s_{t+H_t-1}\}$ from $\tau_i^{skill} = \{s_t, \ldots, s_{t+H_t-1}\}$
  9:         Encode skill: $z \sim q_{\theta_q}(z|\vec{s}_t)$
10:         Decode actions: $\hat{a} = \pi_{\theta_\pi}(a|\vec{s}_t, z)$
11:         Compute prior: $p_{\theta_p}(z|s_t)$
12:         Compute $\mathcal{L}_{\text{embedding}}$ using Eq.3
13:         Sample negative state $s^-$ from a different skill trajectory $\tau_j^{skill} \neq \tau_i^{skill}$
14:         Compute similarity: $f_{\theta_f}(s_t, z, s_{t+b})$ and $f_{\theta_f}(s_t, z, s^-)$
15:         Compute $\mathcal{L}_{\text{contrastive}}$ using Eq.5
16:         Encode states: $h_t = E_{\theta_E}(s_t)$, $h_{t+H_t} = E_{\theta_E}(s_{t+H_t})$
17:         Predict target state: $\hat{h}_{t+H_t} = T_{\theta_T}(h_t, z)$
18:         Decode observation: $\hat{s}_t = O_{\theta_O}(h_t)$
19:         Compute $\mathcal{L}_{\text{target}}$ using Eq.6
20:      **end for**
21:      Compute total loss: $\mathcal{L}_{\text{total}} = \mathcal{L}_{\text{embedding}} + \mathcal{L}_{\text{contrastive}} + \mathcal{L}_{\text{target}}$
22:      Update model parameters $\theta_q, \theta_\pi, \theta_p, \theta_f, \theta_E, \theta_O$, and $\theta_T$ to minimize $\mathcal{L}_{\text{total}}$
23:      **if** $i \mod T_{\text{relabel}} == 0$ **then**
24:         Perform skill length relabeling (Algorithm 2)
25:      **end if**
26: **end for**

---

**Algorithm 2** Skill Length Relabeling

---

**Require:** Dataset $\mathcal{D}$, Similarity function $f$, Threshold $\epsilon$, Min/max skill length $\delta_{\min}, \delta_{\max}$
  1: Sample $N_{\text{sample}}$ states from $\mathcal{D}$
  2: **for** each sampled state $s_t$ **do**
  3:      Retrieve corresponding episode $\tau_{\text{episode}}$
  4:      Encode skill: $z_t \leftarrow q_{\theta_q}(s_t, s_{t+a}, s_{t+b}, s_{t+H_t-1})$
  5:      **for** $\alpha = 1$ to $\text{len}(\tau_{\text{episode}}) - t$ **do**
  6:         Compute similarity: $\text{sim} \leftarrow f_{\theta_f}(s_t, z_t, s_{t+\alpha})$
  7:         **if** $\text{sim} \leq \epsilon$ **then**
  8:            **break**
  9:         **end if**
10:      **end for**
11:      Relabel the skill duration: $H_t' \leftarrow \alpha + 1$
12:      Constrain skill length: $H_t' \leftarrow \max(\delta_{\min}, \min(H_t', \delta_{\max}))$
13: **end for**

---

Here, $r_t$ is the cumulative reward over $H$ -step action sequences, and the KL divergence ensures that the policy samples skills aligned with the learned prior. This improves exploration efficiency by focusing on promising skills, rather than exploring uniformly.

The Soft Actor-Critic (SAC) algorithm is extended to handle skill embeddings instead of primitive actions. The critic updates the Q-values for the selected skills, incorporating the learned skill prior to regularize the policy:

$$Q_\phi(s_t, z_t) = r_t + \gamma Q_\phi(s_{t+H}, \pi_\theta(z_{t+H}|s_{t+H})) \tag{11}$$

The learned skill prior $p_\omega(z|s_t)$ plays a crucial role in guiding the policy toward effective skills during both training and execution, significantly accelerating downstream task learning.

**Model-Based Approach**   In the model-based reinforcement learning setting, we leverage the Skill Dynamics Model from SKiMo (Shi et al. (2022)) to perform efficient long-horizon planning in the skill space. Given an initial state $s_t$ and a skill embedding $z_t$, the skill dynamics model $D_\theta$ predicts the latent state after the skill has been executed for $H$ steps, as $h_{t+H} = T_\theta(h_t, z_t)$, where $h_t = E_\theta(s_t)$ is the latent state produced by the state encoder.

In addition to skill dynamics, the model includes reward prediction and value estimation for each skill. The reward model $R_\phi(h_t, z_t)$ predicts the cumulative reward $r_t$ over the skill's duration, while the Q-value model $Q_\phi(h_t, z_t)$ estimates the expected return from executing the skill. These predictions are used both for planning and for policy optimization, with the Q-value computed as:

$$Q_\phi(h_t, z_t) = r_t + \gamma Q_\phi(h_{t+H}, \pi_\theta(z_{t+H}|h_{t+H})) \tag{12}$$

Temporal Difference Model Predictive Control (TD-MPC) is used for planning. This method leverages skill dynamics and reward models to simulate long-term trajectories, optimizing skill sequences over a defined planning horizon. The Cross-Entropy Method (CEM) is employed to search for the optimal sequence of skills, and the first skill in the sequence is then executed in the environment. The planning step selects skills by maximizing predicted Q-values over the horizon:

$$\text{Plan}(s_t) = \arg \max_{z_{t:t+K}} \sum_{k=0}^{K} [Q_\theta(h_{t+kH}, z_{t+kH})] \tag{13}$$

where $K$ represents the planning horizon. This method allows for efficient skill-based planning by minimizing the number of predictions required, reducing cumulative prediction errors.

The model is trained using a combination of losses, including latent state consistency, reward prediction, and value prediction:

$$L_{\text{model-based}} = \mathbb{E}\left[\lambda_L\|T_\theta(h_t, z_t) - h_{t+H}\|^2 + \lambda_R\|r_t - R_\phi(h_t, z_t)\|^2 + \lambda_V\|Q_\theta(h_t, z_t) - Q_{\text{target}}\|^2\right] \tag{14}$$

where $Q_{\text{target}}$ is the target Q-value computed from future predictions. By using TD-MPC with skill dynamics, reward, and value predictions, this approach enables efficient long-horizon planning in the skill space, improving sample efficiency for reinforcement learning in complex tasks.

### B.3   MODEL IMPLEMENTATION DETAILS

The key models we need to train include the skill encoder $q_{\theta_q}$, skill decoder $\pi_{\theta_\pi}$, skill prior $p_{\theta_p}$, skill-conditioned state representation $\phi_{\theta_\phi}$, state representation $\psi_{\theta_\psi}$, state encoder $E_{\theta_E}$, observation decoder $O_{\theta_O}$, and skill termination predictor $T_{\theta_T}$. Each of these models is implemented as a neural network. The skill encoder $q_{\theta_q}$, skill decoder $\pi_{\theta_\pi}$, skill prior $p_{\theta_p}$, observation decoder $O_{\theta_O}$, and skill termination predictor $T_{\theta_T}$ are all 4-layer MLPs with a hidden dimension of 256, using the ELU activation function. The skill-conditioned state representation $\phi_{\theta_\phi}$ and the state representation $\psi_{\theta_\psi}$ are 2-layer MLPs with a hidden dimension of 128 and employ the ReLU activation function. $\phi_{\theta_\phi}$ and $\psi_{\theta_\psi}$ output the input values to a representation dimension of 16. Additionally, the state encoder $E_{\theta_E}$ is an LSTM with a hidden dimension of 128. The learning rate for all models was set to 0.0003.

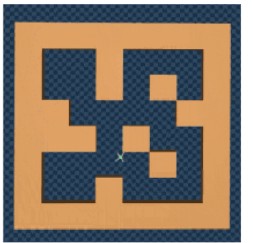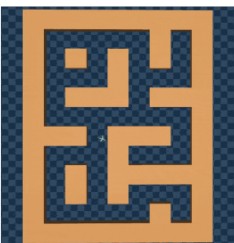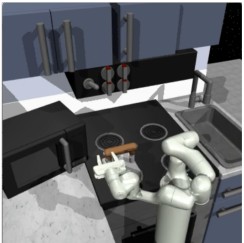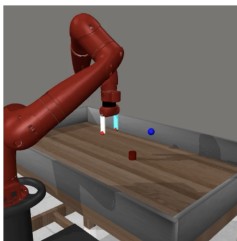

Figure 6: Environments from left-to-right: Antmaze, a long-horizon navigation task with two variants (Antmaze-Medium and Antmaze-Large); Kitchen, where the agent is tasked to complete four independent subtasks; and Meta-world Pick-and-Place, where the goal is to pick up an object and place it at a target location.

## C  EXPERIMENTAL DETAILS

### C.1  ENVIRONMENTS

In this section, we provide a more detailed description of the experimental environments.

The goal of the antmaze task is to guide the ant agent to the goal position within the maze. The observation space has 29 dimensions, capturing the agent's position and joint information. The action space consists of 8 dimensions, corresponding to the control of the agent's 8 joints. In this environment, we set the skill dimension to 5 for all baselines. As mentioned earlier, our framework needs to determine whether the pblueicted skill termination state has been reached. This is assessed using the ant robot's x and y position, the z-coordinate of the torso, its orientation (quaternion x, y, z, w), and the hip joint angles of the 4 legs (1 per leg), resulting in a total of 14 dimensions. The achievement of the skill target state is determined by the L2 distance between the pblueicted state and the current state. The success threshold is set at a distance of 0.5.

In the Kitchen environment, the objective is to manipulate various objects to achieve desiblue states. We used the 'mixed-v0' dataset, where subtasks are presented in a disjoint manner, requiring the agent to generalize across non-sequential task executions. Similar to antmaze, we set the skill dimension to 5 for all baselines. The observation space is 30-dimensional, representing the robot's joint states, while the action space contains 9 dimensions for controlling its joints. Skill termination is determined based on the robot's 9 joint positions and velocities (18 dimensions), with a success threshold distance of 0.1.

For Meta-World's pick-and-place task, the goal is to place an object at a target location. We used datasets that include both successful and failed task executions, as provided by Yoo et al. (2022). The datasets consist of three types: Medium-Replay (MR) with 150 trajectories from intermediate policy stages, Replay (RP) with 100 trajectories spanning the entire learning process (including relevant and irrelevant behaviors), and Medium-Expert (ME) with 50 expert-level trajectories. The observation space is 18-dimensional, representing the robot's joint positions, velocities, and the object's position, while the action space has 4 dimensions. In this environment, the skill dimension is set to 2 for all baselines. The skill termination is evaluated based on the robot's 3D end-effector position and the open/closed state of the gripper (4 dimensions), with a success threshold distance of 0.02.

### C.2  HYPERPARAMETERS

## D  ADDITIONAL EXPERIMENTS

### D.1  RESULTS IN ADDITIONAL ENVIRONMENTS

We conducted additional experiments in an environment that, like Pick-and-Place, allows us to compare results based on data quality. Peg-Insert-Side (Yu et al. (2020)) uses the same robot as Pick-and-Place, but the task involves inserting a peg into a hole in a box. While both tasks involve grasping and moving objects, there are two significant differences between these environments.

Table 3: DCSL hyperparameters.

| Hyperparameter | AntMaze | Kitchen | PickPlace |
|---|---|---|---|
| Batch Size (skill extraction) | 256 | 256 | 128 |
| Batch Size (Downstream learning) | 256 | 256 | 128 |
| Max Global Step (skill extraction) | | 100,000 | |
| Max Global Step (Downstream learning) | | 4,000 | |
| Weighting coefficient $\lambda_{BC}$ | | 2 | |
| Weighting coefficient $\lambda_{SP}$ | | 1 | |
| Weighting coefficient $\lambda_{CL}$ | | 1 | |
| Weighting coefficient $\lambda_{RE}$ | | 1 | |
| Weighting coefficient $\lambda_{ST}$ | | 2 | |
| Skill Dimension | 5 | 5 | 2 |
| Initial Skill Length $H$ | | 10 | |
| Relabeling Interval $T$ | | 20,000 | |
| Relabeling Threshold $\epsilon$ | | 0 | |
| Length Constraint $\delta_{min}$ | | 4 | |
| Length Constraint $\delta_{max}$ | | 30 | |
| Distance Threshold | 0.5 | 0.1 | 0.02 |

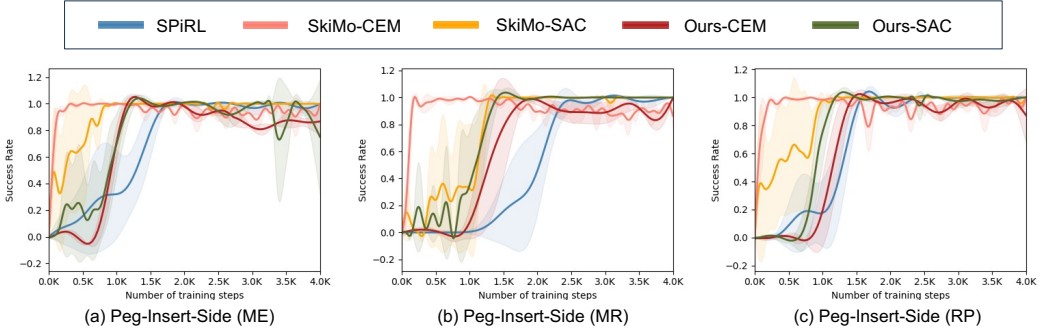

(a) Peg-Insert-Side (ME)    (b) Peg-Insert-Side (MR)    (c) Peg-Insert-Side (RP)

Figure 8: Performance comparison of various methods in the Peg-Insert-Side task across different dataset qualities (ME, MR, RP). All methods demonstrate high success rates, highlighting the task's relative ease compared to Pick-and-Place.

First, there's a difference in the size of the object the robot needs to grasp. In Pick-and-Place, the object is very small, so even a slight misalignment of the robot's gripper can result in a failed grasp. In contrast, the peg in Peg-Insert-Side is longer, making it easier for the robot to grasp.

The second difference is that in Peg-Insert-Side, the robot can use the box as a guide when positioning the object. By moving the peg along the box's surface, the robot can more easily find the hole. Pick-and-Place doesn't offer such guidance, making it a more challenging environment for the robot to complete its task.

These differences are reflected in the experimental results. Regardless of the dataset quality and the method used for finding downstream tasks, all methods showed high success rates in Peg-Insert-Side. This confirms that one of our framework's major strengths - skill length adjustment - becomes more pronounced as the task difficulty increases.

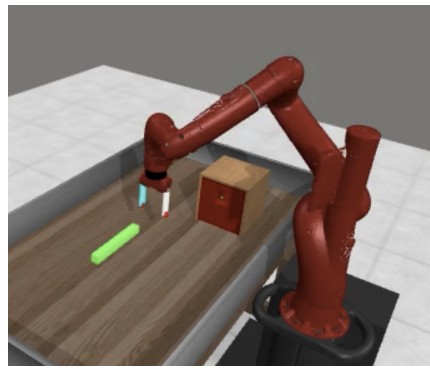

Figure 7: Peg-Insert-Side task: The robot must grasp a long object and insert it into a hole in the box.

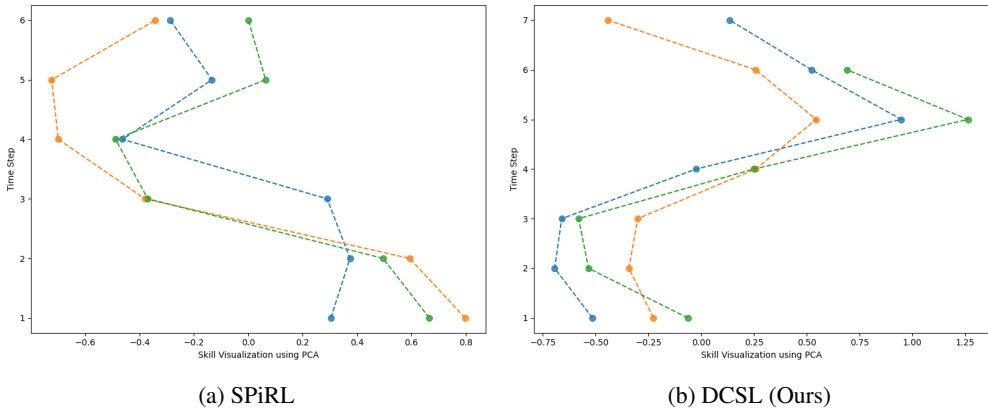

(a) SPiRL                                        (b) DCSL (Ours)

Figure 9: Visualization of skill usage in the pick-and-place task. The x-axis represents the skills reduced through PCA analysis, while the y-axis indicates the timesteps at which the high-level policy operates.

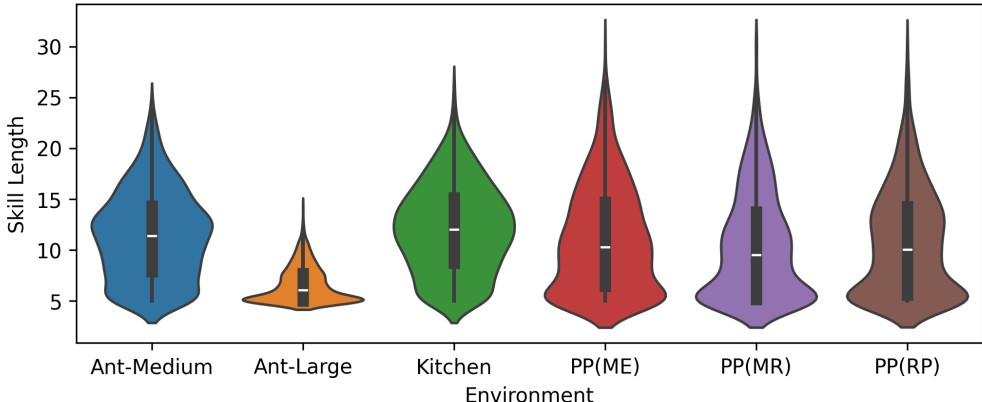

Figure 10: Distribution of relabeled skill lengths across various environments using our framework. These results emphasize the adaptability of DCSL in tailoring skill lengths to the specific demands of each task, improving flexibility and generalization in long-horizon RL tasks.

## D.2 VISUALIZATION OF SKILL SELECTION

We visualized how skills are utilized in the pick-and-place task to verify whether DCSL effectively clusters similar behaviors into similar skills. Using PCA analysis, we reduced the dimensionality of the skills selected by the high-level policy to one dimension. In each episode, the initial and target positions of the object varied. For a more precise analysis, we configured DCSL's high-level policy to sustain a selected skill for the duration of 10 low-level actions, matching the behavior of SPiRL. The results are presented in Fig. 9. These figures illustrate that, despite variations in the object's initial and target positions, the patterns of skill usage in DCSL remain consistent across different scenarios. This consistency suggests that our method captures some level of semantic context in skill utilization, even though further analysis is required for explicit interpretability. In contrast, SPiRL also exhibits instances of selecting similar skills in certain segments. For example, at timestep 2, the skills selected are closely clustered, likely corresponding to the robot arm's gripper performing the action of grasping the object. As previously mentioned, SPiRL embeds action sequences as skills, which explains why similar skills are selected in segments where the gripper's movements are identical. However, outside of these segments, SPiRL demonstrates less consistent patterns of skill usage compared to DCSL.

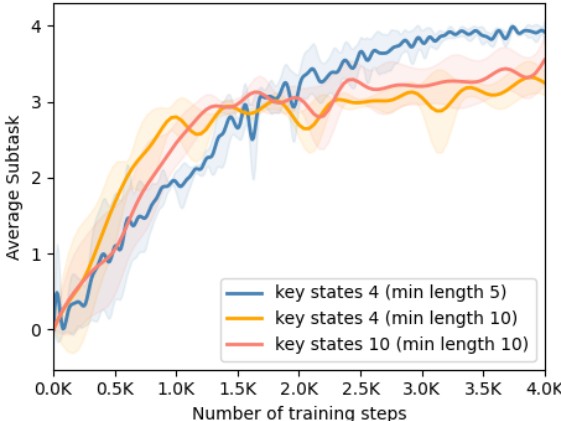

Figure 11: The ablation study on the number of key states for the Kitchen task.

### D.3 Skill Length Analysis

We analyzed the distribution of relabeled skill lengths produced by the DCSL framework across different environments using violin plots (Fig. 10). These plots reveal that skill durations were dynamically adjusted according to state transitions and task complexity, rather than relying on the initial lengths. In the antmaze-medium environment, skill lengths primarily ranged from 10 to 15 steps, reflecting medium-length behaviors suitable for the task. In contrast, skills in the antmaze-large environment were generally shorter, supporting the need for more frequent and precise decision-making. For more complex environments, such as the kitchen and pick-and-place tasks (ME, MR, RP datasets), the skill length distribution was more diverse, indicating that DCSL effectively captured both short and long action sequences necessary to handle varied behaviors. This adaptability highlights the framework's ability to tailor skill durations to the specific demands of each task, enhancing its flexibility and generalization in long-horizon reinforcement learning scenarios.

### D.4 Additional Ablation Studies

#### D.4.1 NUMBER OF KEY STATES

To address the impact of the number of key states on DCSL's performance, we conducted an ablation study in the Kitchen environment. This study explores the relationship between the number of key states and the minimum skill length, which are interconnected parameters in our framework. We compared three configurations: (1) 4 key states with a minimum skill length of 5, (2) 4 key states with a minimum skill length of 10, and (3) 10 key states with a minimum skill length of 10. The results, presented in Fig. 11, show that using 4 key states with a minimum skill length of 5 achieved the highest average return. The superior performance of the 4 key states (min length 5) configuration suggests that allowing for shorter skills can capture more fine-grained behaviors, which is particularly beneficial in manipulation tasks like those in the Kitchen environment. Interestingly, increasing the number of key states from 4 to 10 while maintaining a minimum skill length of 10 showed only a slight improvement in performance. The minimal difference in performance between the configurations with 4 key states (min length 10) and 10 key states (min length 10) can be attributed to the effectiveness of the similarity function in DCSL. This similarity function, learned through contrastive learning, allows the model to capture the semantic context of behaviors regardless of the number of key states used. By focusing on state transitions, DCSL can effectively cluster similar behaviors into coherent skills, even with fewer key states.

#### D.4.2 ABLATION ON SIMILARITY THRESHOLD $\epsilon$

To evaluate the sensitivity of our DCSL framework to the similarity threshold $\epsilon$, we conducted a comprehensive analysis in the pick-and-place environment using the Replay (RP) dataset (Fig. 12.

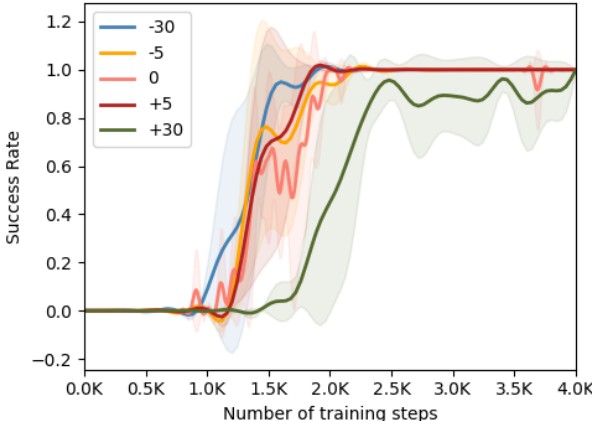

Figure 12: The ablation study on the similarity threshold $\epsilon$ for the pick-and-place task.

Table 4: Comparison of the number of timesteps required to complete pick-and-place.

| Task | w/o Relabeling | with Relabeling |
|---|---|---|
| PP(ME) | **75.1** $\pm$ 16.9 | 80.1 $\pm$ 13.7 |
| PP(MR) | 77.8 $\pm$ 8.1 | **56.1** $\pm$ 5.1 |
| PP(RP) | 98.9 $\pm$ 10.8 | **64.4** $\pm$ 16.3 |

We tested five different $\epsilon$ values: -30, -5, 0, 5, and 30. Our results demonstrated that the framework achieved high success rates across all threshold settings, indicating robustness to this parameter. However, we observed that setting $\epsilon$ to 30 led to a slight increase in the number of training steps required for convergence compared to other values. This can be attributed to the stricter criterion imposed by a higher $\epsilon$ value for considering state transitions as part of the same skill during the length relabeling process. Consequently, this stricter threshold tends to generate shorter skills, which in turn necessitates more training steps for the high-level policy to learn effectively. This analysis provides valuable insights into the trade-off between skill granularity and learning efficiency in our framework, demonstrating its adaptability across different threshold values while highlighting the impact of skill length on downstream task learning.

### D.4.3 COMPARISON OF TASK LEARNING EFFICIENCY WITH AND WITHOUT RELABELING

We conducted a comparison of the timesteps required for task completion across different datasets with skill length adjustment (Table 4). Interestingly, the results revealed that skills extracted from suboptimal datasets enabled tasks to be completed more efficiently compared to those extracted from the high-quality PP(ME) dataset. This finding suggests that, despite containing more irrelevant actions, suboptimal datasets allow DCSL to capture a greater diversity of behaviors as skills, ultimately contributing to more efficient task completion.

## E LIMITATIONS

While DCSL demonstrates promising results across various environments, it also has several limitations that warrant further investigation. First, DCSL shows limited performance with CEM-based planning. Our experiments reveal that the variable skill lengths in DCSL can destabilize the planning horizon in Cross-Entropy Method (CEM)-based approaches. This inconsistency in time scales reduces CEM's effectiveness, particularly in long-horizon tasks, leading to suboptimal performance compared to SAC-based methods. Second, DCSL's generalization to novel tasks remains limited. Although it performs well within the distribution of the training dataset, its ability to transfer learned skills to entirely new tasks or significantly different environments is constrained. This highlights the need for further research into more robust mechanisms for skill transfer. Additionally, combining our

approach with intrinsic rewards that aid exploration could be a viable method for learning new skills through online interaction. Third, DCSL heavily depends on the diversity of the training dataset. Its skill learning is restricted to the behaviors present in the offline dataset, making it incapable of learning skills that are not represented in the training data. This limitation suggests potential future work in incorporating online interaction through exploration rewards to enable the learning of new skills beyond the initial dataset. Fourth, the computational complexity of DCSL is another challenge. The processes of dynamic skill length adjustment and contrastive learning introduce additional computational overhead compared to fixed-length skill learning methods, which could limit its applicability in resource-constrained environments or real-time applications. Finally, there is an issue with the interpretability of the learned skills. While DCSL effectively clusters similar behaviors, the semantic meaning of these clustered skills is not always clear or interpretable. This lack of interpretability could pose challenges in applications where explainability is crucial. Addressing these limitations offers exciting opportunities for future research. Potential directions include developing more robust planning methods compatible with variable-length skills, enhancing skill transfer capabilities, and exploring hybrid offline-online learning approaches to broaden the range of learnable skills.

