# OpenReview forum: "Dynamic Contrastive Skill Learning with State-Transition Based Skill Clustering and Dynamic Length Adjustment"
_ICLR.cc/2025/Conference — ICLR 2025 Poster_

### Official Review · Reviewer_aqwT · 2024-10-30

**Soundness:** 2
**Presentation:** 2
**Contribution:** 2
**Rating:** 5
**Confidence:** 4

**Summary:**

This paper investigates skill learning in reinforcement learning (RL) and proposes Dynamic Contrastive Skill Learning (DCSL), a method to learn skill embeddings based on state transitions. The approach uses contrastive learning to cluster semantically similar skills and dynamically adjusts skill length during the learning process. Experiments are conducted on Ant-Maze, D4RL Kitchen, and Meta-World Pick-and-Place environments, demonstrating that DCSL, combined with the downstream RL algorithm SAC, outperforms previous skill learning methods.

**Strengths:**

* The approach is well-motivated, addressing important challenges in capturing semantically similar skills and adapting skill lengths across different tasks.
* The experiments across multiple benchmarks and tasks provide strong empirical support for the benefits of DCSL over baseline methods.

**Weaknesses:**

* The experimental and ablative studies could be more comprehensive, covering a broader range of tasks and datasets within D4RL and Meta-World.
* The paper’s writing could be clearer and more accessible (see clarification questions below).
* While the paper offers some theoretical insights in Appendix A, DCSL lacks a rigorous theoretical foundation with formal mathematical derivations and analysis.
* The paper does not discuss its limitations, which is essential for understanding the boundaries of its applicability.

**Questions:**

1. **Performance of SPiRL in Figure 4(c)**: The performance of the SPiRL baseline appears significantly worse than reported in the original SPiRL paper on D4RL Kitchen tasks. Could this discrepancy be due to differences in the datasets used in D4RL Kitchen environments?
2. **Definition of training steps (Figure 4\)**: In Figure 4, the x-axis refers to the number of training steps, but RL research typically focuses on environment steps to measure sample efficiency. Could you clarify the meaning of "training steps" in this context? Additionally, would it be possible to provide a comparison of different methods based on sample efficiency?
3. **Additional datasets in D4RL Kitchen**: The experiments in the D4RL Kitchen environment are conducted using only the mixed dataset, but this benchmark includes two additional datasets. Could you extend the comparisons to these datasets? Will DCSL maintain its advantage across different datasets?
4. **Meta-World experiments**: While DCSL performs well on the pick-and-place task in Meta-World, could you test it on more challenging tasks within Meta-World to fully validate its robustness and generalization?
5. **Ablation without similarity function**: In the ablation study, could you include a variant of DCSL that omits the similarity function? This would result in a method that relies purely on state-transition-based skill learning. How would this variant compare to SPiRL, which is action-based skill learning?
6. **Skill target state in continuous space (Line 262\)**: The sentence mentions that a skill target state is considered "reached" during downstream execution, but how do you determine whether a target state is reached in continuous state spaces? Is there a threshold used to make this determination?
7. **Definition of skill trajectory (Line 248\)**: The sentence refers to the "trajectory of skill z." Could you provide a formal definition of what constitutes the trajectory of a given skill?
8. **Negative sampling in contrastive learning (Equation 5\)**: In Equation 5, how is z′≠z determined, given that z lies in a continuous space? For any trajectory other than the anchor trajectory, the skill embedding z’ will always differ from z, so does this mean any state from other trajectories would serve as a negative sample?
9. **Sensitivity to similarity threshold ϵ\\epsilonϵ (Equation 9\)**: How sensitive is DCSL’s performance to the value of the hyperparameter \\epsilon in Equation 9? A sensitivity analysis would be useful to understand the robustness of the method.
10. **Sparse reward problem (Line 369\)**: The paper claims that DCSL addresses the sparse reward problem better, but there is no detailed explanation of the sparse reward issue in the Ant-Maze environments. Moreover, conducting experiments on more tasks with sparse rewards would strengthen this claim.
11. **Generalization to unseen tasks**: Can the skills learned during training be transferred to unseen tasks that are relevant to the training tasks? For example, SPiRL demonstrated generalization by applying learned skills to more complex mazes. Does DCSL exhibit similar generalization capabilities?

---

> ### Author Response · Authors · 2024-11-22
> **Response to Reviewer aqwT (1/2)**
>
> We sincerely thank you for your valuable feedback on our work. Below are our responses to the feedback provided by the reviewer aqwT.
> > **W1: The experimental and ablative studies could be more comprehensive**
>
> - Based on your comments, we conducted additional experiments, and the details will be addressed in the responses to the following questions.
>
> > **W3: DCSL lacks a rigorous theoretical foundation**
>
> - Thank you for your insightful feedback. Based on your suggestion, we have added "Theoretical Analysis of Contrastive Learning for Skill Discrimination" to `Appendix A.1 in the revised paper`.
>
> > **W4: The paper does not discuss its limitations**
>
> - We appreciate your valuable comment. In response, we have included a limitations section in `Appendix E of the revised paper`.
>
> > **Q1: Performance of SPiRL in Figure 4(c)**
>
> - You're correct that there appears to be a discrepancy in SPiRL's performance compared to the original paper. Both our implementation and the original SPiRL used the 'kitchen-mixed' dataset from D4RL. The main difference lies in the skill dimension setting. In our experiments, we set the skill dimension to 5, whereas the original SPiRL paper used 10. This change relates to one of DCSL's key contributions: the ability to cluster similar behaviors into the same skill during the skill extraction process, rather than simply storing action sequences. We chose this smaller skill dimension to evaluate how well DCSL can learn compact skill representations compared to methods like SPiRL. However, this change likely affects SPiRL's performance, as it was originally designed and optimized for a larger skill dimension. We included this explanation in the revision.
>
> > **Q2: Definition of training steps (Figure 4)**
>
> - During the training of DCSL and other baselines, a single training step was conducted after rolling out one episode. Thus, the x-axis in Figure 4 represents the number of episodes. The graph shows results over a maximum of 4000 episode rollouts, demonstrating that DCSL achieves higher sample efficiency compared to other baselines.
>
> > **Q3: Additional datasets in D4RL Kitchen**
>
> - The average returns for each dataset are presented below. Interestingly, SPiRL demonstrates superior performance on datasets other than kitchen-mixed. For kitchen-partial and kitchen-complete, the datasets include trajectories where all target subtasks are successfully completed. In such cases, methods like SPiRL, which store action sequences directly, may overfit to these trajectories, enabling more efficient performance. For kitchen-partial, the presence of diverse and mixed subtasks could have limited the ability of DCSL and SkiMo to effectively learn a sufficiently diverse skill embedding space. In the case of kitchen-complete, where only expert demonstrations are present, the lower performance of DCSL and SkiMo may be attributed to challenges arising from distributional shifts.
>
> | Data    | Ours-SAC    | Ours-CEM    |  SkiMo-SAC    |  SkiMo-CEM    |  SPiRL    |
> |----------|----------|----------|----------|----------|----------|
> | kitchen mixed    | 3.94 &plusmn; 0.01   | 3.00 &plusmn; 0.21  | 3.08 &plusmn; 0.16  | 3.53 &plusmn; 0.11  | 1.85 &plusmn; 0.81  |
> | kitchen partial  | 2.25 &plusmn; 0.22    | 1.79 &plusmn; 0.13   | 2.13 &plusmn; 0.56  | 2.13 &plusmn; 0.13  | 2.86 &plusmn; 0.68  |
> | kitchen complete | 2.42 &plusmn; 0.16   | 2.33 &plusmn; 0.16  | 1.74 &plusmn; 0.04  | 1.99 &plusmn; 0.15  | 2.79 &plusmn; 0.12  |
>
> > **Q4: Meta-World experiments**
>
> - We conducted experiments using the dataset provided in [1], which includes a total of 10 tasks: button-press, door-open, drawer-close, drawer-open, push, reach, window-open, window-close, peg-insert-side, and pick-place. Among these, all tasks except peg-insert-side and pick-place involve relatively simple actions and therefore showed high success rates across all baselines, regardless of dataset quality. Although peg-insert-side requires more complex interactions, its higher success rate across all baselines can be attributed to the larger size of the object and the guidance provided by the robot arm’s movements during interaction. Detailed results and analysis for peg-insert-side can be found in `Appendix D.1`.
>
> [1] Yoo, Minjong, Sangwoo Cho, and Honguk Woo. "Skills regularized task decomposition for multi-task offline reinforcement learning." Advances in Neural Information Processing Systems 35 (2022): 37432-37444.

---

> ### Author Response · Authors · 2024-11-22
> **Response to Reviewer aqwT (2/2)**
>
> > **Q5: Ablation without similarity function**
>
> - We have included an ablation study on the similarity function by modifying `Fig. 4` in the experimental section and adding an analysis of the results. Without the similarity function, skills are represented simply as LSTM embeddings of key states, and length relabeling is not performed. This leads to difficulties in extracting useful skills, resulting in lower performance.
>
> > **Q6: Skill target state in continuous space (Line 262)**
>
> - The method for determining whether the target state has been reached and the corresponding distance thresholds for each environment are detailed in `Appendix C.1`. To summarize, in the antmaze environment, the agent's position and a subset of joint information are used, while in the kitchen and pick-and-place environments, specific joint information of the robot arm is considered. The L2 distance between the current state and the target state is used to assess whether the target state has been reached.
>
> > **Q7: Definition of skill trajectory (Line 248)**
>
> - The trajectory of a skill $z_t$ refers to the sequence of states starting from the initial state $s_t$ and ending at $s_{t+H_t-1}$. Formally, this is defined as $\tau^{\text{skill}}=(s_t, s_{t+1}, \dots, s_{t+H_t-1})$.
>
> > **Q8: Negative sampling in contrastive learning (Equation 5)**
>
> - You raise an important point about negative sampling in continuous skill spaces. Indeed, as z lies in a continuous space, any state not included in the skill trajectory $\tau^{\text{skill}} = (s_t, s_{t+1}, ..., s_{t+{H_t-1}})$ corresponding to z_t can potentially serve as a negative sample from a different skill trajectory. Given that the dataset size is typically much larger than individual skill trajectory lengths, the probability of sampling false negative pairs is low. Even if a few false negative pairs are included, they are unlikely to significantly impact the overall learning direction. This approach ensures a diverse set of negative samples while maintaining the integrity of the contrastive learning process.
>
> > **Q9: Sensitivity to similarity threshold $\epsilon$ (Equation 9)**
>
> - We conducted a sensitivity analysis for the similarity threshold $\epsilon$ and included the results in the appendix. We compared different $\epsilon$ values (-30, -5, 0, 5, 30) in the PP(RP) environment. While high success rates were achieved across all settings, setting $\epsilon$ to 30 required slightly more training steps compared to other values. A higher $\epsilon$ value implies a stricter criterion for considering state transitions as part of the same skill during length relabeling, resulting in shorter skills. Consequently, this leads to the generation of shorter skills, which in turn requires more training steps for the high-level policy to learn effectively.
>
> > **Q10: Sparse reward problem (Line 369)**
>
> - Thank you for bringing this to our attention. We acknowledge that our explanation of the sparse reward problem in the Ant-Maze environments was insufficient. In the Ant-Maze environments, the agent receives a reward of 1 only upon reaching the goal, and 0 otherwise. This sparse reward structure makes exploration a critical factor in successful learning. DCSL's superior performance in these environments can be attributed to its enhanced exploration efficiency. The state-transition based skill representation and dynamic skill length adjustment allow DCSL to capture more meaningful behaviors, leading to more effective exploration in sparse reward settings. Similarly, in the case of the kitchen task, a +1 reward is received only upon completing each subtask, and in the pick-and-place task, the reward is given only when the object is placed at the target location. Therefore, both tasks can be considered sparse reward tasks.
>
> > **Q11: Generalization to unseen tasks**
>
> - Our Kitchen environment experiments demonstrate generalization to unseen tasks, as the agent must perform subtask sequences not present in the training data. However, for environments with structural changes like maze size or layout modifications, additional adjustments may be necessary, similar to SPiRL's approach. DCSL, SPiRL, and SkiMo all rely heavily on skill priors for solving downstream tasks, but these priors can be challenging to compute for previously unseen states. To address this, SPiRL uses local top-down view images around the agent as input, making it invariant to maze size or structure. While this approach could potentially be applied to Antmaze, it may have limitations in capturing the ant agent's joint information from a top-view image alone.

---

> > ### Comment · Reviewer_aqwT · 2024-11-25
> >
> > Thank you for the detailed response. I appreciate the updates on theoretical foundation and ablative studies.

---

> > > ### Author Response · Authors · 2024-11-26
> > > **Thank you for the constructive review.**
> > >
> > > Thank you for your response! Your feedback has greatly helped improve our paper. If you have any additional concerns or questions, please feel free to let us know.

---

### Official Review · Reviewer_pKgb · 2024-11-03

**Soundness:** 3
**Presentation:** 3
**Contribution:** 3
**Rating:** 8
**Confidence:** 3

**Summary:**

The paper proposes DCSL - a method for learning skills based on state sequences from offline data, using a contrastive loss. A similarity function is parameterized by a neural network and learned, which is then used to cluster similar skills together. In addition, the approach also allows for flexible learning skills of various durations by relabelling the duration of each skill based on the same similarity function. The results show better performance on downstream tasks when these skills are used with a model-based or model-free high-level controller.

**Strengths:**

- Skills are more general and less overfitting to specific types of behaviors due to the similarity function.
- Dynamic relabelling allows for skills for different lengths and the paper shows cases where that's beneficial.
- Significantly better performance than baselines on downstream tasks.
- Seems more robust than baselines to lower quality demonstrations in the dataset as shown by the PP ME dataset.

**Weaknesses:**

- Comparisons are only done with variations of two methods (SPiRL and SkiMo), which makes it harder to judge the strength of the method.
- Since the method uses Behavior cloning to learn the skills, I’m not sure how well it will perform in datasets with large amounts of sub-optimal data. While the results show that it outperforms BC-based baselines, there’s no comparison with non-BC methods like offline RL.
- Qualitative comparisons (videos) of the behaviors would be appreciated.

**Questions:**

## Questions
1. The second contribution mentions that the similarity function “clusters semantically similar behaviors into the same skill category.” Just to clarify, is “skill category” used loosely here? As I understand the skills are continuous N dimensional vectors and similar behaviors (in terms of state sequences) will be encoded to similar skills, but there are no discrete groups or categories of skills?


2. Are semantically similar skills close to each other in Z space? For example in Fig. 6c), would behaviors moving in more or less the same direction have similar skill value?


3. The skill prior is used for both the embedding loss and to guide exploration for downstream tasks. How is this skill prior learned (as mentioned in line 687)?


4. Have you done any ablation on the number of key states - is there a specific reason for choosing 4?


5. It is a bit unclear to me how key states are sampled and associated with skills. Each trajectory tau_I contains T state-action pairs.
Does each trajectory have one or several skills associated with it? From line 4 on Algorithm 1 it seems that each trajectory has one skill associated with it and the skill duration H for each trajectory then changes based on the relabelling (Algorithm 2)? What if a single trajectory contains multiple behaviors?

6. How does this method compare to works like OPAL [1] or to using unsupervised skill discovery with offline RL to learn skills from a dataset  [2], [3]? If possible, a comparison could strengthen the contributions.

## Review summary:
Overall I think this is a high quality work and the proposed method is promising. The paper is well-written and easy to follow (with a couple of caveats mentioned above). The main concern is that I think more comparisons should be made with other relevant methods which learn skills from offline datasets with or without behavior cloning.

References:

[1]	A. Ajay, A. Kumar, P. Agrawal, S. Levine, and O. Nachum, “OPAL: Offline Primitive Discovery for Accelerating Offline Reinforcement Learning,” May 04, 2021, arXiv: arXiv:2010.13611.

[2]	S. Park, T. Kreiman, and S. Levine, “Foundation Policies with Hilbert Representations,” presented at the Forty-first International Conference on Machine Learning, Jun. 2024.

[3]	J. Kim, S. Park, and S. Levine, “Unsupervised-to-Online Reinforcement Learning,” Aug. 27, 2024, arXiv: arXiv:2408.14785. doi: 10.48550/arXiv.2408.14785.

---

> ### Author Response · Authors · 2024-11-22
> **Response to Reviewer pKgb (1/2)**
>
> We sincerely thank you for your valuable feedback on our work. Below are our responses to the feedback provided by the reviewer pKgb.
> > **W1 & W2: Comparisons are only done with variations of two methods**
>
> - Thank you for the valuable feedback. In response, we have included comparisons with additional methods that also utilize offline datasets.
>
>
> | Task    | DCSL (SAC)    |  BC    | CQL    | CQL+Off-DADS    |  CQL+OPAL    |
> |----------|----------|----------|----------|----------|----------|
> | Ant-medium    | 68.0 &plusmn; 36.9    |  0.0    | 53.7 &plusmn; 6.1   | 59.6 &plusmn; 2.9 | 81.1 &plusmn; 3.1  |
> | Ant-large     | 73.7 &plusmn; 5.9     |  0.0    | 14.9 &plusmn; 3.2   | -                 | 70.3 &plusmn; 2.9  |
> | Kitchen-mixed | 94.7 &plusmn; 1.48    |  47.5   | 52.4 &plusmn; 2.5   | -                 |  69.3 &plusmn; 2.7 |
>
> > **W3: Qualitative comparisons of the behaviors would be appreciated.**
>
> - Thank you for the suggestion. Since skills in DCSL are represented as continuous N-dimensional vectors, direct visual comparison of individual skills through agent behaviors can be challenging. However, we have addressed this by providing qualitative evaluations in both navigation and manipulation tasks. Beyond the visualization of skill behaviors in antmaze (Figure 6), we conducted additional qualitative analyses in the pick-and-place task. As shown in `Appendix D.2 in the revised paper (Figure 9)`, even when the initial and target positions of the object vary, the skill patterns exhibit consistency, demonstrating the robustness of the learned skills across different scenarios.
>
> > **Q1: Is “skill category” used loosely here?**
>
> - Thank you for the insightful comment. You are correct that in our framework, skills are represented as continuous N-dimensional vectors and are not classified into discrete groups or categories. To improve clarity, we will revise the phrase “clusters semantically similar behaviors into the same skill category” to “clusters semantically similar behaviors into similar skill embeddings,” as this more accurately reflects the nature of our approach. This change will be included in the revision.
>
> > **Q2: Are semantically similar skills close to each other in Z space? & Would behaviors moving in more or less the same direction have similar skill value?**
>
> - The analysis of semantically similar skills was addressed in response to W3.
> - DCSL does not explicitly learn the value of each skill but instead focuses on extracting skills from the dataset. As observed in `Fig. 5c in the revised paper`, the short-distance movements reflect the extraction of stopping behaviors from the dataset, likely due to the agent encountering specific stopping points. Additionally, as shown in `Fig. 6 in the revised paper`, these stopping locations coincide with walls, suggesting that the agent has learned to either stop or transition to another skill upon reaching such positions.
>
> > **Q3: How is this skill prior learned?**
>
> - The skill prior is learned through the second term in `Equation 3`. Specifically, it is trained by minimizing the KL divergence between the output distribution of the skill encoder and the skill prior. This ensures that the learned skill prior effectively captures the distribution of likely skills based on the dataset.
>
> > **Q4: Have you done any ablation on the number of key states**
>
> - Thank you for the excellent suggestion. We chose to use a fixed number of key states to uniformly process training data, regardless of skill length. Since the minimum skill length was set to 5, we selected 4 key states to ensure compatibility with shorter skills. We also hypothesized that the similarity function used for skill discrimination would mitigate the impact of variance from a smaller number of key states. To validate this, we conducted an ablation study in the kitchen environment, setting the minimum skill length to 10 and comparing the performance with 4 and 10 key states. As shown in `Appendix D.4.1 in the revised paper` and `Figure 11`, using 10 key states slightly improved performance compared to 4, but the difference was minimal, demonstrating the robustness of DCSL to the number of key states.
>
> |     |  key states 4 (min length 5)   | key states 4 (min length 10)    | key states 10 (min length 10)    |
> |----------|----------|----------|----------|
> | Average Return    | 3.99 &plusmn; 0.03   | 3.32 &plusmn; 0.22  | 3.55 &plusmn; 0.33  |

---

> > ### Author Response · Authors · 2024-11-22
> > **Response to Reviewer pKgb (2/2)**
> >
> > > **Q5: It is a bit unclear to me how key states are sampled and associated with skills.**
> >
> > - Thank you for pointing this out. The distinction between the dataset trajectory $\tau$ and the skill trajectory $\tau^\text{skill}$ used in skill learning was not made clear, which may have caused confusion. The initial skill length H_t is assigned to every state-action pair $(s_t, a_t)$ in the dataset. The skill trajectory $\tau^\text{skill}$, which is used for training, is defined as the sequence from $(s_t, a_t)$ to $(s_{t+H_t-1}, a_{t+H_t-1})$. By setting $\tau^\text{skill}$ to be shorter than $\tau$, our method can extract diverse skills from a trajectory $\tau$ that contains a variety of behaviors. Based on your feedback, we revised the algorithm to better clarify this distinction.
> >
> > > **Q6: How does this method compare to works like OPAL?**
> >
> > - The additional experiments addressing this point were covered under W1. Thanks to your suggestion, we were able to compare DCSL against a broader range of baselines, including OPAL, which further highlights and strengthens the contributions of our method.

---

> > ### Comment · Reviewer_pKgb · 2024-11-26
> >
> > Thank you for the detailed response. I am mostly happy with the changes made during the rebuttal, especially the additional ablations and comparison, which showcase the advantages of the proposed method. I do have one small clarifying question:
> >
> >  It's still not fully clear how you learn the skill prior (original Q3):
> > > Q3: How is this skill prior learned?
> > The skill prior is learned through the second term in Equation 3. Specifically, it is trained by minimizing the KL divergence between the output distribution of the skill encoder and the skill prior. This ensures that the learned skill prior effectively captures the distribution of likely skills based on the dataset.
> >
> > Should this refer to Equation 4 in the updated manuscript? If so, as I understand it this trains the skill embedding network $q(z|s)$ based on an action reconstruction loss (the $\lambda_{BC}$ term) while trying to stay close to the prior $p(z|s)$, but how is the prior $p(z|s)$ learned in the first place? I see this as a sort of VAE loss with a learned prior, is this right? Are you optimizing **both** the encoder *and* the skill prior jointly on the KL divergence loss? I think specifying the trainable parameters of each component might make it clearer.

---

> > > ### Author Response · Authors · 2024-11-28
> > > **Response to Reviewer pKgb**
> > >
> > > Thank you for your response. Your feedback has been invaluable in improving our paper. I apologize for the confusion in my previous response. You are correct that I should have referred to `Equation 4`. Below is our response to your question regarding the skill prior.
> > >
> > > > **Q: How to learn skill prior.**
> > >
> > > - To address your question, we modified `Equation 3` to provide a clearer explanation of how the skill prior and skill encoder are learned. (Due to page limitations, we replaced the original `Equation 1` with a textual explanation.) `Equation 3` can be broadly divided into two parts. The first part includes the VAE components: the $\lambda_{\text{BC}}$ term and the $\beta$ term. These terms facilitate the learning of the skill encoder and skill decoder through action reconstruction and regularization toward a tanh-transformed standard Gaussian distribution $p(z)$. The second part is the $\lambda_\text{SP}$ term, where the stop gradient operator (sg) blocks the gradient of $q(z|\vec{s})$, ensuring that only $p(z|s)$ is updated. This approach allows the skill prior to adapt to the distribution of embeddings generated by the encoder without affecting the encoder itself. It promotes consistency between the prior and the encoder while preserving their distinct roles in the skill learning process. Therefore, while both the skill encoder and skill prior are continuously trained, it would be inaccurate to describe them as being jointly optimized. This method of learning the skill encoder, skill decoder, and skill prior is also similarly employed in SPiRL [1] and SkiMo [2]. Following your suggestion, we have explicitly incorporated the trainable parameters into the `equations in the revised paper` and reflected these updates in `Appendices A and B in the revised paper`.
> > >
> > >
> > > We appreciate your careful review and suggestions. If you have any additional questions, please feel free to let us know.
> > >
> > > [1] Pertsch, Karl, Youngwoon Lee, and Joseph Lim. "Accelerating reinforcement learning with learned skill priors." Conference on robot learning. PMLR, 2021.
> > >
> > > [2] Shi, Lucy Xiaoyang, Joseph J. Lim, and Youngwoon Lee. "Skill-based model-based reinforcement learning." arXiv preprint arXiv:2207.07560 (2022).

---

> > > > ### Comment · Reviewer_pKgb · 2024-11-28
> > > >
> > > > Thank you for the further clarification, I think it is much clearer now in the revised version.

---

> > > > > ### Author Response · Authors · 2024-11-29
> > > > > **Thank you for the constructive review.**
> > > > >
> > > > > Thank you for your response! We sincerely thank you once again for taking the time to review our paper and provide feedback. Your efforts have greatly contributed to improving our paper.

---

### Official Review · Reviewer_fLcu · 2024-11-04

**Soundness:** 3
**Presentation:** 2
**Contribution:** 3
**Rating:** 6
**Confidence:** 4

**Summary:**

Use discriminability to identify skills, then use a skill similarity function trained using contrastive learning with a state-skill embedding and a state embedding. The representation is then used as an input to the termination condition, which compares the current autoencoded state to the skill embedding. The model is then trained as a joint objective. Compute skill duration as the maximum number of steps where the skill similarity is greater than a threshold value.

**Strengths:**

Investigates some of the core questions in skill learning, including variable-length skills and skill differentiability.

Offline skill learning is an increasingly appplicable field, especially as large robotics datasets become more prevalent

Provides a mostly clear and understandable description of a complex method and motivation for each algorithmic choice.

**Weaknesses:**

Of the original claims made in the introduction, there is a disconnect between those and the empricial results. In particular, while it is clear that the skills appear to be more effective for downstream tasks the original claims do not appear to be verified. It is not as obvious that the skills capture ``more semantic context'', as the only experiment in this context is antmaze (Figure 6), and while this appears to show some separation, it is not particularly semantic. It would be more useful to provide some clearly semantic task transfer, such as opening multiple different drawers in franka kitchen with the same skill. Similarly, it is not clear the how the skill lengths are used for performance, in particular the effect of skill length relabeling appears to be marginally significant. It would be valuable to have some analysis indicating that for some tasks shorter/longer skills are assigned appropriately. Finally, it is not actually that obvious why this method takes advantage of the offline RL setting and could not be directly applied to unsupervised skill learning.

The method itself is straightforward, but there are a substantial number of models that must be trained together (two encoders in the similarity measure, and encoder-decoder in the state encoder, target state model, skill prior, policy). Training these together appears to have been done using a few objectives without any hyperparameters. In practice, combining these kinds of models often requires some kind of tuning, which seems like it would be a limitation. How is this done in practice for this method?

In the formulation of the skill embeddings, two randomly sampled intermediate states are used to identify the skill embedding. This seems like it could introduce a significant amount of variance. Also, it is not clear how that number of intermediate states is decided. A more careful analysis of this design choice seems appropriate (is this something used by related work? How does a different number of intermediate states compare? What are the theoretical ramifications of such a choice?) and is missing from the paper.
The level of comparison is a little bit limited. In particular, this method seems at least peripherially related to unsupervised skill learning work, so it seems appropriate to make at least some comparison to that work. I expect that without access to the advantage of offline data those methods would not perform particularly well, but in a domain like antmaze, they may exihibit better skill differentiation.

As mentioned before, a more in-depth analysis of skill length would be valuable in this work. Even in the domains where additional skill length showed performance benefit, it is not obvious why it would, since these are pick and place tasks. Some kind of analysis which indicated how much time was wasted without skill length relabeling would probably be demonstrative.

**Questions:**

See Weaknesses

---

> ### Author Response · Authors · 2024-11-22
> **Response to Reviewer fLcu (1/2)**
>
> We sincerely thank you for your valuable feedback on our work. Below are our responses to the feedback provided by the reviewer fLcu.
> > **W1: It's not clear that the skills capture "more semantic context" as claimed.**
>
> - As you pointed out, the 'semantic' meaning of each skill we extracted may not be entirely clear. We acknowledge that a more rigorous demonstration of semantic context would be beneficial. While using advanced models like LLMs to interpret skill semantics is beyond the scope of this work and could be considered for future research, we have attempted to visualize the clustering of similar behaviors in the embedding space. Specifically, in `Appendix D.2 in the revised paper`, we've included graphs showing how skills are utilized in pick-and-place tasks. These figures demonstrate that despite varying initial and target object positions, the patterns of skill usage remain similar across different scenarios. This similarity in skill utilization patterns suggests that our method is capturing some level of semantic context, even if it's not explicitly interpretable without further analysis.
>
> > **W2: It would be valuable to have some analysis indicating that for some tasks shorter/longer skills are assigned appropriately.**
>
> - The analysis of how skill lengths vary across different environments is addressed in `Appendix D.3 in the revised paper`, "Skill Length Analysis," with supporting visuals in `Figure 10`. To summarize, in the antmaze environments, the relabeled skill lengths clustered within specific ranges: 10–15 for antmaze-medium and 5–10 for antmaze-large. This reflects the task-specific nature of the skill lengths, with shorter skills being more suitable for the larger and more complex antmaze. In contrast, more complex tasks like kitchen and pick-and-place exhibited a broader distribution of skill lengths, demonstrating DCSL's adaptability in tailoring skill durations to the diverse requirements of these tasks.
>
> > **W3: It's not obvious why this method specifically takes advantage of the offline RL setting and couldn't be applied to unsupervised skill learning.**
>
> - Thank you for the insightful comment. Methods like DADS [1] in unsupervised skill learning typically rely on intrinsic rewards to facilitate exploration and learn new skills. In contrast, our approach, which extracts skills from offline datasets, does not incorporate exploration into the skill learning process, making it less directly applicable to unsupervised skill learning frameworks. However, our proposed skill similarity function could potentially be combined with intrinsic rewards to support exploration, which we agree is an interesting direction for future research. We will include this consideration in the limitation section to highlight the potential extension of our method to unsupervised settings.
>
> > **W4: The method involves training multiple models together without clear explanation of hyperparameter tuning.**
>
> - Thank you for pointing this out. We have addressed this concern by adding weights to the terms in Equations 4, 6, and 7 to clarify the hyperparameter tuning process.
>
> > **W5: The choice of the number of intermediate states needs more justification and analysis.**
>
> - Thank you for raising this point. We chose to use a fixed number of four key states to uniformly process training data regardless of skill length. Since the minimum skill length was set to 5, selecting four key states ensured compatibility with shorter skills. Additionally, we hypothesized that the skill similarity function would mitigate the variance introduced by a smaller number of key states, maintaining robust skill discrimination. To validate this, we conducted experiments in the kitchen environment, setting the minimum skill length to 10 and comparing results using 4 and 10 key states. As shown in `Appendix D.4.1 in the revised paper` and `Figure 11`, using 10 key states slightly improved performance compared to 4, but the difference was negligible, demonstrating DCSL's robustness to the choice of key state count.
>
> |     |  key states 4 (min length 5)   | key states 4 (min length 10)    | key states 10 (min length 10)    |
> |----------|----------|----------|----------|
> | Average Return    | 3.99 &plusmn; 0.03   | 3.32 &plusmn; 0.22  | 3.55 &plusmn; 0.33  |
>
>
> [1] Sharma, Archit, et al. "Dynamics-aware unsupervised discovery of skills." arXiv preprint arXiv:1907.01657 (2019).

---

> ### Author Response · Authors · 2024-11-22
> **Response to Reviewer fLcu (2/2)**
>
> > **W6: The level of comparison with other methods is limited.**
>
> - Thank you for your observation. In response, we expanded our comparisons to include additional baselines. For unsupervised skill learning, we initially conducted experiments with DADS [1] in the antmaze environment, but the results were insignificant. Therefore, we replaced it with experiments using an offline variant, CQL+off-DADS [2], which yielded more meaningful results.
>
> | Task    | DCSL (SAC)    |  BC    | CQL    | CQL+Off-DADS    |  CQL+OPAL    |
> |----------|----------|----------|----------|----------|----------|
> | Ant-medium    | 68.0 &plusmn; 36.9    |  0.0    | 53.7 &plusmn; 6.1   | 59.6 &plusmn; 2.9 | 81.1 &plusmn; 3.1  |
> | Ant-large     | 73.7 &plusmn; 5.9     |  0.0    | 14.9 &plusmn; 3.2   | -                 | 70.3 &plusmn; 2.9  |
> | Kitchen-mixed | 94.7 &plusmn; 1.48    |  47.5   | 52.4 &plusmn; 2.5   | -                 |  69.3 &plusmn; 2.7 |
>
> > **W7: A more in-depth analysis of skill length would be valuable in this work.**
>
> - Thank you for highlighting this. To illustrate, in the pick-and-place task, expert demonstrations consist solely of picking an object and placing it at the target location in every episode. In such cases, fixed skill lengths can still yield useful skills. However, in suboptimal datasets, actions necessary for task completion are mixed with irrelevant behaviors, such as failing to interact with the object or dropping it after picking it up. In these cases, extracting common behaviors as skills and appropriately adjusting skill lengths becomes crucial. The impact of skill length adjustment on task success rates is discussed in the experimental section. As you suggested, we also compared the timesteps required for task completion across datasets with skill length adjustment. Interestingly, we found that skills extracted from suboptimal datasets performed tasks more efficiently than those from the high-quality PP(ME) dataset. This indicates that while suboptimal datasets contain more irrelevant actions, DCSL captures a greater diversity of behaviors as skills, ultimately leading to more efficient task completion.
>
> | Task    |  w/o Relabeling   | with Relabeling    |
> |----------|----------|----------|
> | PP(ME)    | 75.1 &plusmn; 16.9   | 80.1 &plusmn; 13.7  |
> | PP(MR)    | 77.8 &plusmn; 8.1    | 56.1 &plusmn; 5.1   |
> | PP(RP)    | 98.9 &plusmn; 10.8   | 64.4 &plusmn; 16.3  |
>
> [2] Sharma, Archit, et al. "Emergent real-world robotic skills via unsupervised off-policy reinforcement learning." arXiv preprint arXiv:2004.12974 (2020).

---

> > ### Comment · Reviewer_fLcu · 2024-11-25
> > **Response to Authors**
> >
> > I appreciate the additional experiments, and I believe they improve the quality of the paper. I will raise my score accordingly as I believe the work is sufficient to be accepted.

---

> > > ### Author Response · Authors · 2024-11-26
> > > **Thank you for the constructive review.**
> > >
> > > Thank you for your response! We truly appreciate the time you took to review our paper and for raising your score. Your feedback has greatly helped us improve our work.

---

### Author Response · Authors · 2024-11-22
**General Response**

We sincerely thank all the reviewers for their valuable feedback. Your comments have helped us address the weaknesses in our paper and significantly improve its overall quality. In this section, we provide a summary of the revisions made based on your suggestions.
> **Paper revision summary**

All revisions have been highlighted in blue text.
- (For reviewer pKgb) We revised “same skill category” to “similar skill embeddings” to improve clarity.
- (For reviewer fLcu) We added weighting coefficients for each model in the equations.
- (For reviewer fLcu, pKgb) We included additional baselines that utilize offline datasets.
- (For reviewer aqwT) We added an ablation study on the similarity function in Fig. 4.
- (For reviewer aqwT) We included a theoretical analysis of "Contrastive Learning for Skill Discrimination" in Appendix A.1.
- (For reviewer pKgb) We revised the term related to the skill trajectory in Algorithm 1.
- (For reviewer fLcu, pKgb) We added a qualitative evaluation of skills in the pick-and-place task in Appendix D.2.
- We revised Fig. 10 to address an issue in the original graph, which included skill lengths assigned to states at the end of episodes that were not used for skill learning.
- (For reviewer fLcu, pKgb) We added an ablation study on the number of key states in Appendix D.4.1.
- (For reviewer aqwT) We added an ablation study on the similarity threshold in Appendix D.4.2.
- (For reviewer fLcu) We added an experiment on the impact of relabeling on the timesteps required to solve tasks in Appendix D.4.3.
- (For reviewer aqwT) We added a limitations section in Appendix E.

---

### Meta-Review · Area_Chair_6gG5 · 2024-12-18

**Metareview:**

The paper proposes a method for learning skills from offline data using constrastive methods. The reviewers agree that the core ideas in the paper are quite interesting and results to back the ideas up. There are several minor concerns on the limited difficulty of the tasks and the quality of writing. Both I believe can be addressed before the camera ready deadline.

**Additional Comments On Reviewer Discussion:**

After discussion reviewer aqwT increased their score and hence we have several reviewers rating this work positively.

---

### Decision · Program_Chairs · 2025-01-22

Accept (Poster)